# 🏛 ChartMuseum: Testing Visual Reasoning Capabilities of Large Vision-Language Models

**Liyan Tang** **Grace Kim** **Xinyu Zhao** **Thom Lake** **Wenxuan Ding**
**Fangcong Yin** **Prasann Singhal** **Manya Wadhwa** **Zeyu Leo Liu** **Zayne Sprague**
**Ramya Namuduri** **Bodun Hu** **Juan Diego Rodriguez** **Puyuan Peng** **Greg Durrett**

The University of Texas at Austin
{lytang, yeeunk, xinyuzhao}@utexas.edu
https://chartmuseum-leaderboard.github.io

## Abstract

Chart understanding presents a unique challenge for large vision-language models (LVLMs), as it requires the integration of sophisticated textual and visual reasoning capabilities. However, current LVLMs exhibit a notable imbalance between these skills, falling short on visual reasoning that is difficult to perform in text. We conduct a case study using a synthetic dataset solvable only through visual reasoning and show that model performance degrades significantly with increasing visual complexity, while human performance remains robust. We then introduce CHART-MUSEUM, a new Chart Question Answering (QA) benchmark containing 1,162 expert-annotated questions spanning multiple reasoning types, curated from real-world charts across 184 sources, specifically built to evaluate complex visual and textual reasoning. Unlike prior chart understanding benchmarks—where frontier models perform similarly and near saturation—our benchmark exposes a substantial gap between model and human performance, while effectively differentiating model capabilities: although humans achieve 93% accuracy, the best-performing model Gemini-2.5-Pro attains only 63.0%, and the leading open-source LVLM Qwen2.5-VL-72B-Instruct achieves only 38.5%. Moreover, on questions requiring primarily visual reasoning, *all* models experience a 35%-55% performance drop from text-reasoning-heavy question performance. Lastly, our qualitative error analysis reveals specific categories of visual reasoning that are challenging for current LVLMs.

## 1 Introduction

While a substantial body of work on foundation model reasoning has focused on math and code [5, 7, 10, 13, 8, 27, 33], multimodal reasoning remains understudied despite its unique challenges, such as the representation bottleneck of the visual encoder [38]. To address this, recent multimodal benchmarks test capabilities such as solving math problems that involve visual components [22, 39] or overall performance with domain knowledge [35, 43].

However, there is a distinction between multimodal problems that admit solutions with textual reasoning and those that *require* visual reasoning. For instance, a geometry problem can be formalized into a symbolic representation to derive a solution without any visual reasoning. Meanwhile, problems like face recognition have been shown to defy description in language [32], with techniques like chain of thought actually causing degradation in model accuracy [20].

Chart understanding represents an ideal domain to explore the spectrum of textual and visual reasoning. Charts are designed to present data in ways that enable a viewer to quickly derive insights that are not obvious from raw data. Answering questions about charts blends visual interpretation, extraction

39th Conference on Neural Information Processing Systems (NeurIPS 2025) Track on Datasets and Benchmarks.

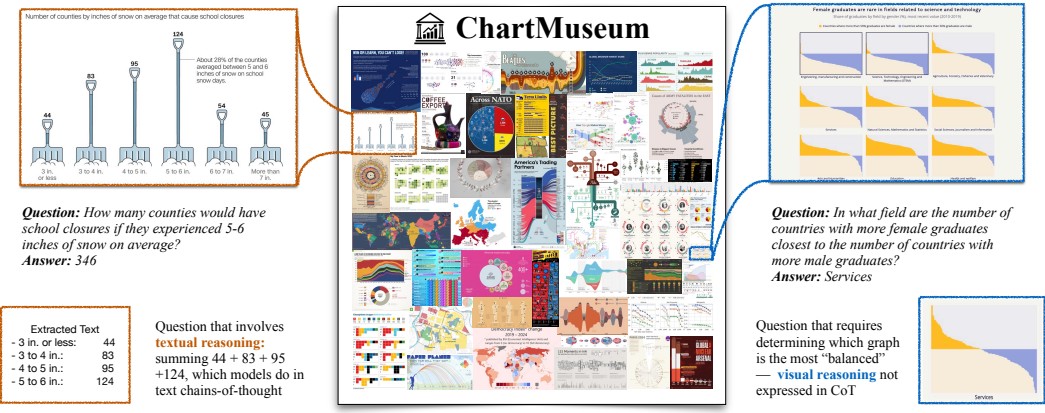

Figure 1: CHARTMUSEUM contains a broad collection of charts with associated questions and answers, designed to test LVLMs at both textual and visual reasoning capabilities.

of textual information, and reasoning in natural language. Yet we show that existing chart question answering datasets often prioritize textual reasoning (Section 2.2) or have limited real-world chart sources (Table 2), limiting the scope of their evaluation. Despite high accuracy on these benchmarks, our case study with synthetic charts in Section 2.3 shows that LVLMs still fall short at purely *visual* reasoning, even in settings where humans achieve nearly perfect accuracy.

To address these gaps, we introduce CHARTMUSEUM, a comprehensive chart question-answering (QA) dataset collected to evaluate LVLMs on complex visual and textual reasoning over realistic charts (Section 3). CHARTMUSEUM was created by 13 computer science researchers and consists of 1,162 *(image, question, answer)* tuples sourced from 928 unique real-world images across 184 websites. Unlike prior benchmarks (e.g., ChartBench [42], CharXiv [40], ChartQAPro [24]) where questions are typically model-generated and later refined by annotators—potentially limiting their realism and diversity—all questions in CHARTMUSEUM were curated by researchers without assistance from LLMs. Each question underwent a manual multi-stage review process to ensure question quality and answer objectivity. Figure 1 shows the types of charts and reasoning skills highlighted in the dataset; more question-answer examples can be seen in Appendix A.2.

We evaluate 10 recent open-source models and 11 proprietary models on ChartMuseum. Our benchmark shows clear disparities across model families and scales: the best open model Qwen2.5-VL-72B achieves 38.5% accuracy while Gemini-2.5-Pro reaches 63.0%, both below human performance (93.0%). We further divide questions into four fine-grained reasoning types, showing that model performance on questions that require complex visual reasoning is 35%-55% worse than on questions that only require complex textual reasoning. Our error analysis (Section 5) identifies the major visual task categories represented by our dataset, and finds that the best proprietary models over-rely on textual reasoning when asked to perform visual reasoning tasks, and continue to struggle with visual comparisons, picking out objects based on visual markers, and reasoning about line trajectories.

Our contributions include: (1) An analysis of existing chart understanding datasets and a new synthetic experiment showing that past work has under-evaluated visual reasoning; (2) A new dataset, CHARTMUSEUM, containing diverse, high-quality human-curated questions that are challenging for frontier LVLMs; (3) Qualitative error analysis to identify shortcomings of recent LVLMs for future research.

## 2   Background and Motivation

Recent work [23, 42, 40, 9, 41, 24, 12] has annotated datasets of reasoning-based questions for chart understanding tasks. However, they often overlook a critical distinction: whether the reasoning relies on textual or visual information. In this section, we formalize the difference between *visual reasoning* and *textual reasoning* in the context of the chart understanding task.

## 2.1 Terminology

**Visual Reasoning**    We define visual reasoning as drawing inferences from a chart which are *primarily visual* and arise more naturally from a visual comparison than from reasoning in natural language (Figure 1). This involves making inferences based on *primarily visual* aspects of the chart; i.e., where interpretation of the chart's graphical relationships is essential and more expedient than reasoning via natural language. For instance, reasoning that two variables are highly correlated in a scatterplot is visual; there is no practical way to express this in natural language or math short of extracting many values and approximating a computation of correlation. Reasoning that a slice of a pie chart with the label "37%" corresponds to 37% is *not* visual, as the insight is expressed in text in the chart. As a middle ground, reasoning that one bar in a bar graph is higher than another bar is also visual, but it could also be also inferred via value extraction into text. In this case, the visualization is designed to support this comparison easily, whereas extraction of the values is more cumbersome.

We define *visual extraction* as a subclass of visual reasoning consisting of cases where numeric information is extracted through visual interpretation. For instance, reasoning that a bar has a value of around 37 by comparing it with the y-axis labels is visual extraction, as is extracting values from a heatmap by decoding against the legend. In these cases, substantial visual interpretation is required to infer the values, differing from the labeled-value case of the pie chart discussed above.

**Textual Reasoning**    We define two cases of textual reasoning. The first case is reasoning about extracted information using natural language, including logical, arithmetic, and comparative operations over the information that has been extracted and verbalized in the model's chain of thought. Such analysis could be performed by a text-only LLM. The second case is direct extraction of text from the chart (e.g., numeric annotations of bar/line graph values, legend labels, or axis titles) into token space. While this process involves reasoning about visual data, such abilities are very narrow and we consider images of text to be fundamentally textual in nature.

## 2.2 Prior Chart Understanding Benchmarks Over-represent Textual Reasoning

To understand the extent to which existing *real-image* based chart understanding benchmarks rely on textual reasoning, we conduct an experiment on the widely used chart QA benchmark, ChartQA [23], where a model takes as input a chart image and a question, and then predict the answer. Specifically, for each image in ChartQA, we prompt Claude-3.7-Sonnet [2] to extract all *explicit text* information (e.g. title, caption, and annotated values) from the corresponding chart, *without* exposing it to the question. Only text that is explicitly present in the chart is captured. If a chart is unannotated (i.e., the underlying data are shown only visually), such values are not extracted. The model is explicitly instructed not to perform any visual extraction (e.g., inferring values that aren't explicitly given) beyond color extraction, and we observe that it follows the instructions reliably. We then provide this extracted information to Claude-3.7-Sonnet and evaluate its ability to answer the question **without taking the images as input**. The extraction prompt and an example of *(chart, question, extracted text)* pair can be found in Appendix Figure 23 and 24.

As shown in Table 1, Claude-3.7-Sonnet achieves 74.1% when relying solely on explicit textual information, only around 13% lower than their performance when directly using chart images (87.4%). We also conduct the same experiment on our benchmark CHART-MUSEUM and the results shows that relying solely on extracted information yield much worse performance (15.2% compared to 61.3%). The 46% performance gap in CHARTMUSEUM **reflects information that is inherently visual and cannot be captured in the easily-accessible text form**.

| Dataset | Extraction | Image |
|---|---|---|
| ChartQA | 74.1 | 87.4 |
| CHARTMUSEUM | 15.2 | 61.3 |

Table 1: Accuracy (%) of Claude-3.7-Sonnet on ChartQA and CHARTMUSEUM (Section 3) when provided either extracted textual information or images.

Our human study finds that even if the extracted text is accurate and complete in CHARTMUSEUM, humans cannot get sufficient information to answer the question with only the extracted text. Overall, those results suggest that **this current widely used benchmark does not strongly test LVLMs' visual reasoning capabilities.**

## 2.3 Visual Reasoning: Case Study on Synthetic Data

**A Synthetic Testbed for Visual Reasoning**    Our analysis raises the question of how well models can do at reasoning when text extraction is insufficient. To investigate this, we probe visual reasoning capabilities using a synthetic dataset consisting of 13 different visual reasoning questions across 5

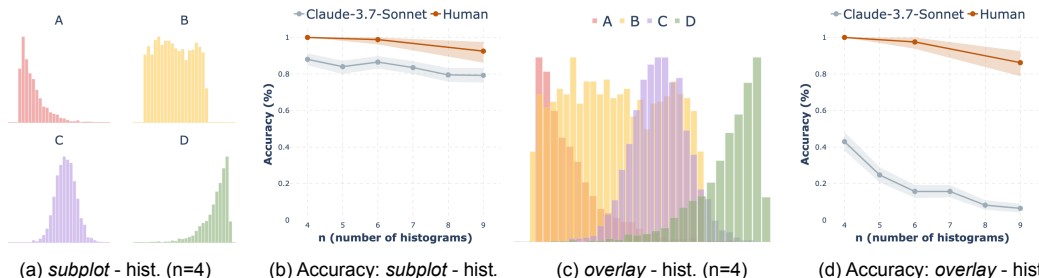

**Question (subplot a, c)**: *Compare figures in the plot and list all figures that are left-skewed (having long-tails to the left).*

(a) *subplot* - hist. (n=4)     (b) Accuracy: *subplot* - hist.     (c) *overlay* - hist. (n=4)     (d) Accuracy: *overlay* - hist.

Figure 2: Visual reasoning case study over histograms. We show the subplot and overlay setups on histograms in (a) and (c) with $n = 4$. Results on Claude-3.7-Sonnet and humans over values of $n \in [4, 9]$ are shown in (b) and (d). Complete questions can be found in Table 8.

different chart types: density, histogram, line, scatter, and violin. Critically, these plots do not feature any written content and cannot be solved through text extraction.

In the synthetic dataset, we introduce two hypothetical yet realistic scenarios commonly encountered in complex real-world chart data: *Overlay* and *Subplot*. In the *Overlay* setup, $n$ figures are layered on top of each other within a single image, capturing the idea of high visual complexity. The *Subplot* setup displays each figure in its own distinct subplot within the same image, capturing the idea of visual correspondence across a figure. This subplot format aligns with prior work on multi-hop visual reasoning, such as MultiChartQA [45]. In both settings, we test for $n \in [3, 9]$. More details of our dataset generation process and questions can be found in Appendix B.1.

**Experiment and Results**    For each question category, we test Claude-3.7-Sonnet on 100 examples and compute the accuracy by averaging across all questions within the same chart category and applying bootstrapping to estimate confidence intervals. We also evaluate human performance on this task with a human study conducted by a subset of the authors; details can be found in Appendix B.1.

We show the performance on *histogram* in Figure 2 and the results for other categories in Appendix B.1. Here we observe a steady decline in model performance in both overlay and subplot setup as $n$ increases. However, we observe that increased visual complexity doesn't significantly affect humans' visual reasoning. This suggests that **when LVLMs cannot rely on text extraction shortcuts, their visual reasoning capabilities are severely limited and degrade with complexity.** Hence, future chart reasoning dataset can incorporate questions with higher visual complexity to test VLMs in a new way, which we proceed to do in Section 3.

## 3   CHARTMUSEUM

CHARTMUSEUM is a chart QA benchmark designed to evaluate reasoning capabilities of LVLMs over real-world chart images. The benchmark consists of 1162 *(image, question, short answer)* tuples and exclusively targets at questions that requires non-trivial textual and visual reasoning skills. A comparison between our benchmark and existing representative chart QA benchmarks can be found in Table 2. The dataset is collectively annotated by a team of 13 researchers in computer science.

### 3.1   Data Annotation Requirements

**Image Source Collection**    Annotators collectively discussed potential image sources, which were then distributed among the annotators. Each annotator was then responsible for finding and annotating images from their assigned image sources. In total, we had 184 distinct website domains, including academic-style charts (such as those from arXiv papers), infographics (e.g., from Reddit and Tableau), and creative or unconventional charts sourced from various websites. To measure the diversity of our dataset, we include benchmark statistics in Figure 3 and compare our chart distribution against recent real-world chart benchmarks.

**Question and Answer Annotation Guidelines**    We established two main requirements for question annotation to ensure high-quality, evaluable questions. First, we require a **large answer space**, explicitly avoiding binary questions or simple comparisons between two entities. Instead, annotators

|  | ChartBench | ChartQA | CharXiv | ChartQAPro | CHARTMUSEUM |
|---|---|---|---|---|---|
| Multiple Chart Sources | ✗ | ✓ | ✗ | ✓ | ✓ |
| Real-World Charts | ✗ | ✓ | ✓ | ✓ | ✓ |
| Entirely Human-written Questions | ✗ | ✓ | ✓ | ✗ | ✓ |
| Wide Range of SOTA Model Acc. | ✗ | ✗ | ✓ | ✗ | ✓ |

Table 2: **Comparison between CHARTMUSEUM and existing chart understanding benchmarks in Chart QA.** Questions in our benchmark are manually curated *without* assistance from LLMs. There is a wide range of accuracy performance across the most recent LVLMs on our benchmark and the reasoning set of CharXiv, which only contains charts from arXiv papers.

were instructed to formulate questions where the answer space has at least 4 options. Second, we maintain strict **objectivity in answers**: all questions in CHARTMUSEUM have unambiguous, objectively verifiable answers. This differs from recent benchmarks [23, 42, 41] that allow for approximate numerical answers within a margin of error, which can compromise evaluation reliability across questions, as some questions may require exact answers. For charts with unannotated data (e.g., the visual reasoning question in Figure 1), we focus exclusively on comparative questions that yield unique answers without requiring tolerance margins. **Note that we do not use LLMs in assisting question creation.**[1]

**Excluded Question Types**    To ensure question quality and evaluability, we also excluded several question types. We avoid "why" and "how" questions, as these typically yield lengthy, potentially subjective responses that are difficult to evaluate objectively. Descriptive questions that merely ask about visually apparent information were also excluded, as all reasoning questions implicitly require such descriptive understanding. Additionally, we omit joint or compound questions that combine multiple queries (*e.g., "What is the sum of the highest value in subplot A and the lowest value in subplot B?"*), as these are unrealistic.

**Question Classification**    Since making inferences from charts often involves both visual and textual reasoning. We classify all chart understanding questions into the following four categories using the reasoning type defined in Section 2.1:

• *Textual Reasoning Questions* can be solved almost exclusively with textual reasoning;

• *Visual Reasoning Questions* are most easily answerable from visual aspects of the chart;

• *Text/Visual Reasoning Questions* can be answered by either primarily text or primarily visual reasoning;

• *Synthesis Reasoning Questions*, require both textual and visual reasoning.

During annotations, we instructed annotators to classify each question into one of the four predefined question categories. Examples of each question category can be found in Appendix A.

### 3.2   Annotation and Quality Check

Each data point in the final benchmark underwent the following annotation process: (1) selecting a high-quality and interesting chart; (2) manually creating a question-answer pair for the image; (3) quality reviewing by the first author of this work who was not among the 13 annotators; and (4) iteratively refining the annotation through discussion with the annotators to improve question clarity and naturalness, answer correctness and objectivity, as well as consensus on the question categories.

The annotation process consisted of a practice session followed by two formal annotation sessions. In the practice session, each annotator annotated 10 *(image, question, short answer)* tuples, which were then reviewed by the independent reviewer to calibrate the annotator's approach with the requirements outlined in Section 3.1. In the following two formal annotation sessions, each annotator created an additional 40 and 50 examples, respectively. The reviewer verified answer correctness

---

[1]At an early stage of the project, we explored the use of templated questions, a common strategy used in existing work, to help with annotation, but were not satisfied with the distinctiveness or quality of generated questions. We aim to generate questions that are unique to each different chart. When not using templates, our annotators generally pose questions targeting the specific core messages that the chart aims to convey, which requires understanding and interpretation of the chart.

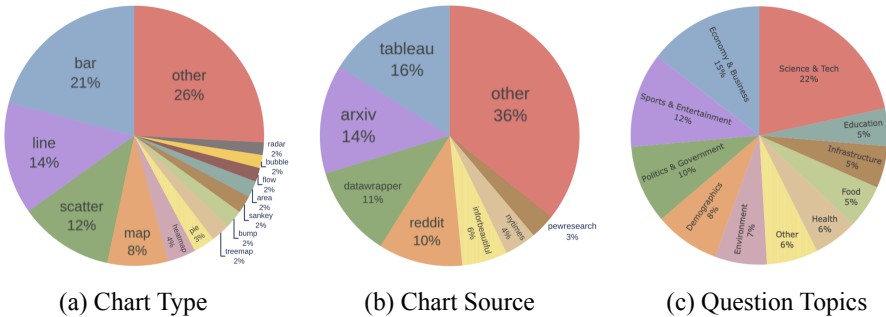

| (a) Chart Type | (b) Chart Source | (c) Question Topics |

Figure 3: The composition of CHARTMUSEUM: (a) depicts the distribution of chart types; (b) presents the sources from which these charts were collected; (c) displays the major question topics found in the dataset, tagged using Claude-3.7-Sonnet.

by independently answering the created questions; any disagreements were resolved via discussions between the reviewer and the annotator.

On average, each *(image, question, short answer)* tuple required 20 minutes of total effort: 10 minutes for chart selection and the initial question-answer pair annotation, 5 minutes for quality review and feedback, and 5 minutes for iterative refinement. This process resulted in approximately 400 total annotation hours for the complete CHARTMUSEUM benchmark with 1,162 examples.

We create a dev/test split with 162 and 1000 examples, respectively, ensuring there are no shared images between the two sets. Unless otherwise stated, we report model performance on the test set.

## 4 Experiments

### 4.1 Model Comparison

We benchmark the performance of state-of-the-art LVLMs for chart question understanding. For proprietary models, we include (1) **GPT-4o**, **GPT-4.1-mini**, **GPT-4.1**, **o3**, **o4-mini** [1, 28, 29] from OpenAI; (2) **Claude-3.5-Sonnet**, **Claude-3.7-Sonnet** [2] from Anthropic; and (3) **Gemini-1.5-Flash/Pro** and **Gemini-2.5-Pro** [19] from Google. For open-source models, we include (1) **Qwen2.5-VL-3B/7B/32B/72B-Instruct** [4] from Alibaba; (2) **InternVL3-2B/8B/38B/78B** [6] from Shanghai AI Lab; and (3) **Pixtral-Large-Instruct** [36] from Mistral AI. We also include the latest specialized chart understanding model **Bespoke-MiniChart-7B** [34] from Bespoke Labs. We use the chain-of-thought prompt in Appendix Figure 26. More model details can be found in Appendix D.

**Human Performance**  To evaluate human performance on CHARTMUSEUM, we conducted a small-scale annotation study with six annotators divided into two groups of three. For each group, we sampled five examples from the annotations of each of the remaining ten annotators (*i.e.*, excluding the three in the group), resulting in a set of 50 examples per group. In total, we collected annotations for $50 \times 2 = 100$ examples, with each example independently labeled by three annotators. The majority-vote human performance is reported in Table 3, while individual annotator results are provided in the Appendix A.

**Evaluation Metric**  All questions in CHARTMUSEUM have unique answers since (1) we avoid questions requiring numeric answers from unannotated chart elements, and (2) our multi-stage review process ensures all answers are objective and unambiguous. We use LLM-as-a-Judge as the main evaluation method to account for paraphrases, using the prompt in Appendix Figure 27.

### 4.2 Results

**CHARTMUSEUM reveals a wide range of model performance.**  We highlight the best performing models across proprietary models and open-source models with varying sizes in Table 3.[2] Unlike

---

[2]We conducted a paired bootstrap test on the performance difference between the best model Gemini-2.5-Pro and all other models on the overall benchmark and 4 reasoning subsets. We found that Gemini-2.5-Pro outperformed every model for which the absolute difference was >3% with p-value < 0.05. In general, differences of about 3-4% on the dataset hold up under bootstrap tests, making it large enough to differentiate practically meaningful differences in model performance.

| CHARTMUSEUM | Visual (510) | Synthesis (133) | Visual/Text (234) | Text (123) | Overall (1000) |
|---|---|---|---|---|---|
| *Open-Source Models (2B and 3B)* | | | | | |
| InternVL3-2B | 12.2 | 13.5 | 18.4 | 30.1 | 16.0 |
| Qwen2.5-VL-3B | 16.7 | 21.1 | 26.5 | 28.5 | 21.0 |
| *Open-Source/Specialized Models (7B and 8B)* | | | | | |
| Qwen2.5-VL-7B | 19.4 | 24.8 | 36.3 | 41.5 | 26.8 |
| InternVL3-8B | 23.5 | 24.8 | 32.9 | 42.3 | 28.2 |
| Bespoke-MiniChart-7B | 26.3 | 32.3 | 41.0 | 54.5 | 34.0 |
| *Open-Source Models (32B and more)* | | | | | |
| InternVL3-38B | 26.3 | 30.8 | 35.0 | 52.0 | 32.1 |
| InternVL3-78B | 26.9 | 34.6 | 41.0 | 59.3 | 35.2 |
| Qwen2.5-VL-32B | 29.0 | 36.1 | 46.2 | 62.6 | 38.1 |
| Pixtral-Large-124B | 31.6 | 36.1 | 40.6 | 65.9 | 38.5 |
| Qwen2.5-VL-72B | 30.4 | 35.3 | 42.3 | 68.3 | 38.5 |
| *Proprietary Models* | | | | | |
| Gemini-1.5-Flash | 22.7 | 30.8 | 36.3 | 56.1 | 31.1 |
| Gemini-1.5-Pro | 31.0 | 43.6 | 49.6 | 65.9 | 41.3 |
| GPT-4o | 31.8 | 45.1 | 50.9 | 65.9 | 42.2 |
| GPT-4.1 | 37.1 | 53.4 | 54.3 | 78.9 | 48.4 |
| GPT-4.1-mini | 43.9 | 48.1 | 59.8 | 80.5 | 52.7 |
| Claude-3.5-Sonnet | 45.7 | 53.4 | 61.5 | 78.0 | 54.4 |
| Claude-3.7-Sonnet | 50.6 | 55.6 | 69.2 | 88.6 | 60.3 |
| *Proprietary Models - Reasoning* | | | | | |
| o3 (high) | 50.4 | 63.2 | 69.7 | 85.4 | 60.9 |
| o4-mini (high) | 51.2 | 66.2 | 68.4 | 86.2 | 61.5 |
| Claude-3.7-Sonnet (think) | 52.5 | 56.4 | 71.8 | 86.2 | 61.7 |
| Gemini-2.5-Pro | 53.3 | 64.7 | 70.1 | 87.8 | 63.0 |

Table 3: Accuracy performance comparison of models on the test set of the CHARTMUSEUM benchmark. **Humans achieve an overall accuracy of 93.0% on a randomly sampled subset consisting 100 examples.** Results show a detailed breakdown across different reasoning types alongside the unweighted overall accuracy. We highlight the highest performance in each of the three model categories. *Key findings: (1) Proprietary models significantly outperform open-source models; (2) Visual reasoning remains weaker than textual reasoning across all LVLMs; (3) A large margin exists between the best model and human performance.*

previous widely evaluated benchmarks such as ChartQA [23] where model accuracies clustered tightly between 85% and 90% (Appendix A.1), our benchmark shows a 24.5% accuracy gap between the best open-source model Qwen2.5-VL-72B-Instruct (38.5%) and the best proprietary model Gemini-2.5-Pro (63.0%) in our benchmark. The specialized chart-understanding model Bespoke-MiniChart-7B, while surpassing other open-source 7B models by a large margin and approaching 72B model performance, still falls far behind proprietary models, highlighting the need for stronger specialized chart understanding models. Finally, human performance (93.0%) exceeds the best proprietary and open-source models by 30.0% and 54.5%, respectively, emphasizing the large room for improvement in chart understanding.

**Visual reasoning performance lags 35% to 55% behind textual reasoning and falls far short of near-perfect human visual reasoning.** Consistent with our findings on ChartQA (Section 2.2), models generally perform the best on questions that rely heavily on textual reasoning (*i.e.* the "Text" column in Table 3). When faced with questions that mainly require complex visual reasoning (*i.e.* the "Visual" column), performance drops significantly. Models such as GPT-4.1, Qwen2.5-VL-72B, and Bespoke-MiniChart-7B show performance decrease of more than 50% on the visual reasoning subset compared to the textual reasoning subset. While the performance degradation is less pronounced for models like Claude-3.7-Sonnet, o3 (high), and Gemini-2.5-Pro, these still show approximately

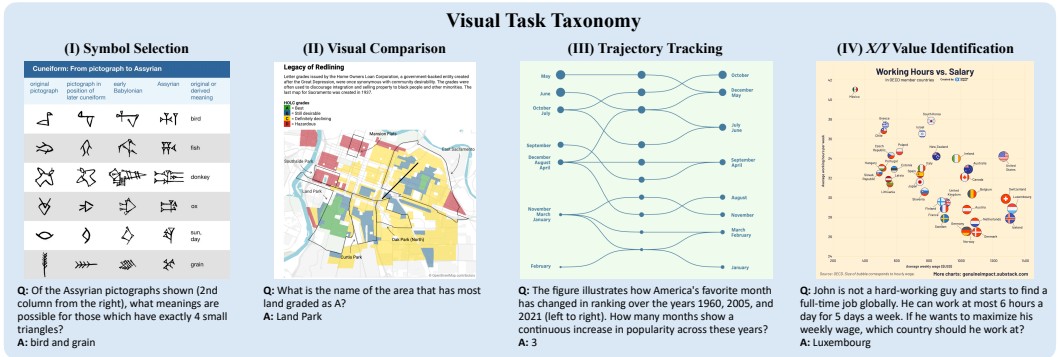

Figure 4: The four categories of visual reasoning tasks that we identify and use for error categorization.

35% absolute accuracy drops, highlighting a persistent shortcoming in visual reasoning. While these questions are exceedingly hard for models, humans achieve near perfect performance on the sampled visual reasoning set (56/57 correct, or 98.2%).

**Reasoning models yield minimal improvements.** Although recent work [8, 19, 3] has shown that LLMs can perform significantly better with extended thinking (i.e., lengthy chain-of-thought with strategies including planning, self-reflection, and self-verification) on tasks such as math [10, 22], and code [15, 30], we do not observe this trend in chart understanding. The improved performance of all reasoning models is within 3% of Claude-3.7-Sonnet without extended thinking. In fact, Claude-3.7-Sonnet with extended thinking (61.7%) achieves only a 1.4% improvement over its standard version (60.3%) and even demonstrates decreased performance over several question categories. In Section 5, we observe that this limited improvement stems primarily from fundamental limitations in visual reasoning capabilities.

## 5 Qualitative Analysis

While it is clear that visual reasoning questions are difficult for models, this blanket term does not sufficiently diagnose model errors. We come up with a taxonomy of skill categories used in ChartMuseum questions (Section 5.1), and examine models' visual reasoning errors to identify skill shortcomings (Section 5.2).

### 5.1 Visual Task Taxonomy

We sample 50 random examples from ChartMuseum (excluding *Textual Reasoning Questions*) to identify the most common visual tasks among questions in our dataset. We find four broad categories of visual tasks and present an example of each in Figure 4:

- **Symbol Selection:** Identifying objects in the chart that match a specific visual criteria such as legend color, shape, pattern, or outline.

- **Visual Comparison:** Comparing multiple objects (or groups of objects) based on their size, height, spatial placement, color intensity (as in a heatmap), or range (as with clusters).

- **Trajectory Tracking and Judgment:** Tracking the position of an element represented by a line or arrow (e.g. in a line chart, across a map, or in a graph) and describing its attributes (e.g. movement direction and slope) or relationship to another visual element.

- **X/Y Value Identification:** Identifying the locations or values of chart elements. To avoid ambiguity, our questions only ask for exact $x$-/$y$- values as an answer when they clearly correspond to a labeled tick mark.

We show the percent of questions within our sample that involve each of these skills in Table 4. A more detailed breakdown of specific visual comparison skills can be found in Table 9, and additional examples of each task type can be found in Appendix C.1.

### 5.2 Visual Reasoning Errors

Answering visual reasoning questions requires composing various skills from Section 5.1 and sometimes there are multiple ways to solve a given problem, although with different difficulties. We

| Task Involved | % |
|---|---|
| Symbol Selection | 20% |
| Visual Comparison | 50% |
| Trajectory Tracking | 26% |
| X/Y Value Identification | 22% |

Table 4: Percent of randomly sampled ($N = 50$) ChartMuseum (excluding *Textual Reasoning*) questions that involve a task from each category. A question can involve tasks from multiple categories.

| Error Type | Claude-3.7 | Gemini-2.5-Pro |
|---|---|---|
| Symbol Selection | 34% | 28% |
| Visual Comparison | 28% | 26% |
| Trajectory Tracking | 14% | 12% |
| X/Y Value Identification | 6% | 28% |
| Strategy Error | 16% | 2% |
| Textual Reasoning Error | 6% | 2% |

Table 5: Percent of sampled error instances ($N = 50$ for each model, excluding *Textual Reasoning* questions) with each error type. Note that a model can make more than one type of error on a question, and even for primarily visual questions, a *Textual Reasoning Error* can still occur for reasons such as question misinterpretation or arithmetic errors.

analyze 100 random error instances (50 each for Claude-3.7-Sonnet and Gemini-2.5-Pro) excluding *Textual Reasoning* questions, and present an error breakdown in Table 5. Even among the labeled *Visual Reasoning*, *Visual/Text*, and *Synthesis* questions, a *Text Reasoning Error* may occur for reasons such as question misinterpretation, incorrect text extraction, or arithmetic errors. However, these occur very infrequently (6% of Claude-3.7-Sonnet and 2% of Gemini-2.5-Pro error instances examined), and the vast majority of errors are due to failures in visual reasoning tasks as defined in Section 5.1.

For many visual comparison tasks in our benchmark, extracting exact $x/y$-values is not necessary or expected. A model may still choose to do so in its chain of thought, and downstream computation using incorrect extracted values can lead to an incorrect final answer. In some cases, the exact values cannot even be correctly judged by a human. For these cases, we define a *Strategy Error* as follows:

- **Strategy Error** A model misses the intended visual reasoning "trick" that is required to solve the question, and instead resorts to a divergent chain-of-thought (often involving extracting explicit $x/y$-values or giving up entirely). This usually occurs when the value of a desired element is not explicitly stated, but is implied relative to other visual elements. Examples can be found in Appendix C.2, Figures 20, 21, and 22.

We find that Claude-3.7-Sonnet has a high proportion of explicit strategy errors. *X/Y Value Identification Errors* are overrepresented for Gemini-2.5-Pro compared to the proportion of visual skills identified across the dataset in Table 4. Most occur when the model struggles with identifying the correct x-tick label corresponding with a bar or annotated point on a graph. A few (3/14) of these errors are made on questions actually intended to assess visual comparison skills and can be straightforwardly answered by comparing the size or height of visual objects (e.g. bars on a graph), but the model chooses to approximate their values for comparison and does so incorrectly, indicating over-reliance on textual reasoning strategies.

## 6 Related Work

The evolution of chart understanding benchmarks under the framework of question answering shows a clear trajectory towards greater realism, complexity, and diversity. Early benchmarks [17, 26, 16] relied on synthetic chart images and template-generated questions. Benchmarks have become more realistic, including ChartQA [23], which introduced human-annotated questions alongside real charts. Further work developed more challenging benchmarks such as CharXiv [40] and ChartQAPro [24]; however, CharXiv draws from a narrow source (arXiv papers), while ChartQAPro does not effectively distinguish performance of frontier models (Appendix A.1). Additionally, both benchmarks use model-generated questions with human review.

Recently we have seen models [8, 29, 19, 3, 4] making significant progress in reasoning capabilities. However, they have been focusing on textual domains such as math and coding [5, 13], and even some visual reasoning benchmarks focus on math problem solving [22, 39]. Many non-math multi-modal benchmarks [21, 43, 35] mainly rely or evaluate on models' domain knowledge and overall capability, not specifically targeting visual reasoning. Evaluating LVLMs is important as they face inherent limitations on visual reasoning, including weak vision encoders [18], misalignment in decoding visual features [11], limited abstract visual reasoning skills [14] and failures to identify textually

describable features [38]. While [37] has broadly shown that LVLMs' visual reasoning lags behind textual reasoning, our benchmark provides a focused investigation into chart understanding, explicitly isolating visual and textual reasoning skills to reveal where and how models fail.

# 7 Conclusion

We present CHARTMUSEUM, a high-quality, human-curated benchmark for chart understanding based on real-world images, designed to evaluate LVLMs on complex textual and visual reasoning tasks across diverse chart types. We show that there is a wide range of accuracy performance across open and proprietary models, yet all fall far short of human performance. More importantly, their visual reasoning capabilities are particularly weak—performing 35%-55% worse than their text-based reasoning. Our qualitative analysis shows that this is due to their limitations in handling different visual reasoning tasks. We hope that CHARTMUSEUM can be a reliable testbed for future LVLM development in strong reasoning across both modalities.

**Limitations** Our benchmark is limited to charts and questions in English, which may not reflect performance in multilingual settings. However, since most current LVLMs are optimized for English, this focus provides a timely evaluation of their capabilities. Second, the benchmark focuses on question answering with short answers, excluding other chart understanding tasks like summarization or open-ended responses. We argue that short-answer QA is an effective proxy for identifying model weaknesses, as other tasks can often be reformulated as QA or are inherently subjective to evaluate (*e.g.,* summarization). Finally, our benchmark does not include unanswerable questions, as we prioritize evaluating models' ability to answer questions where ground-truth answers exist.

# Acknowledgments

This work was partially funded by Good Systems,[3] a UT Austin Grand Challenge to develop responsible AI technologies. This work was additionally supported by the National Science Foundation under Cooperative Agreement 2421782 and Simons Foundation (NSF-Simons AI Institute for Cosmic Origins – CosmicAI, https://www.cosmicai.org/), a grant from Open Philanthropy, by NSF CAREER Award IIS-2145280, and by the NSF AI Institute for Foundations of Machine Learning (IFML). This research has been supported by computing support on the Vista GPU Cluster through the Center for Generative AI (CGAI) and the Texas Advanced Computing Center (TACC) at the University of Texas at Austin. This research was supported in part with funding from the Defense Advanced Research Projects Agency's (DARPA) SciFy program (Agreement No. HR00112520300). The views expressed are those of the author and do not reflect the official policy or position of the Department of Defense or the U.S. Government.

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

## A CHARTMUSEUM Details

**Benchmark Statistics** We show the statistics of CHARTMUSEUM over charts, questions, and answers. The result is in Figure 6. Following [40], we use GPT-4o as the tokenizer to measure the length of questions and answers.

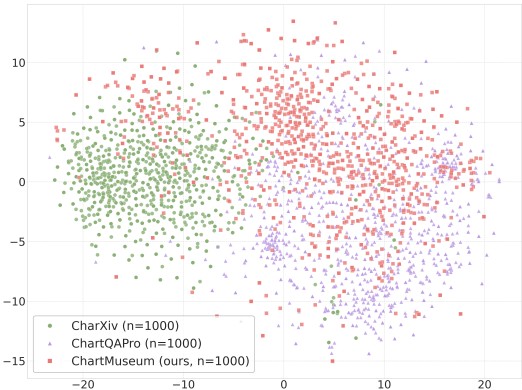

Figure 5: We show a t-SNE visualization, comparing CLIP encodings [31] from 1000 randomly sampled charts in CHARTMUSEUM against those from several real-image based chart benchmarks.

| Statistics | Value |
|---|---|
| ***Dataset Split*** | |
| dev / test | 162 / 1000 |
| ***Charts*** | |
| # charts | 1162 |
| # unique charts | 928 |
| average size (px) | $1590 \times 1243$ |
| ***Questions*** | |
| # questions | 1162 |
| # unique questions | 1161 |
| # unique tokens | 4824 |
| maximum length | 176 |
| average length | 26.7 |
| ***Answers*** | |
| # unique tokens | 1582 |
| maximum length | 23 |
| average length | 2.9 |

Figure 6: Statistics of CHARTMUSEUM.

**Comparison of Chart Type Distribution Against CharXiv**   We randomly sampled 50 arXiv charts from CHARTMUSEUM and CharXiv [40], respectively, and manually checked the chart type distributions. The result is shown in Table 6. Among the sampled charts, 58% from CHARTMUSEUM and 54% from CharXiv contain subplots.

| Type | Percentage |
|---|---|
| Line | 26% |
| Bar/Histogram | 22% |
| Multiple chart types | 16% |
| Scatter | 12% |
| Box plot | 6% |
| Area | 6% |
| Flowchart | 4% |
| Heatmap | 2% |
| 3D chart | 2% |
| Density | 2% |
| Radar | 2% |

| Type | Percentage |
|---|---|
| Line | 40% |
| Scatter | 16% |
| Multiple chart types | 16% |
| Bar/Histogram | 14% |
| Heatmap | 14% |

Table 6: Comparison of chart type distributions from CHARTMUSEUM (left) and CharXiv (right) for 50 randomly sampled arXiv charts each.

**Analysis of Question Diversity**   To evaluate the diversity of questions in CHARTMUSEUM, we conducted a comparative analysis with CharXiv. We randomly sampled 50 questions from each dataset and find that that 25/50 questions in CharXiv and 26/50 in CHARTMUSEUM focused on extreme values (e.g., highest, lowest). Of the sampled 50 questions from CharXiv, the questions share similar reasoning required to answer the questions, but mainly differ by the fact that the questions are based on different chart images (Figure 10 from CharXiv). We believe this is the fundamental limitation of creating question templates and letting models generate similar questions based on the template. In CHARTMUSEUM, we directly ask humans to independently write questions with no shared prior. Even if many questions are about extreme values, the questions are not direct lookups, and require strong multi-step reasoning capability unique to each image.

**Individual Human Performance**   In Section 4, we show that humans achieve a performance of 93% based on the majority vote. Here we report individual human performance: 76%, 90%, 92%, 94%, 96%, 96%.

**Model Performance Over the Development Set**    We report the model performance over the dev set in Table 11. Since there are only 162 examples in total, we only report the overall accuracy performance.

**Benchmark License**    Our benchmark is licensed under CC BY-SA 4.0. Copyright of all included charts is retained by their original authors and sources.

## A.1    A Comparison Against Existing Chart QA Benchmarks

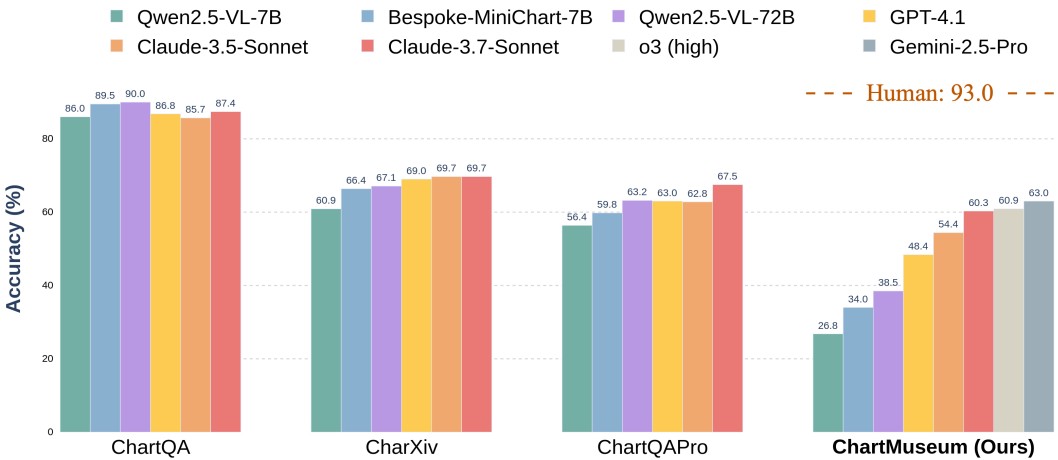

Figure 7: Model performance across existing chart understanding benchmarks and our benchmark CHARTMUSEUM. Our benchmark separates model performance by large margins and recent models falls far behind human performance.

We show a comparison between our benchmark and existing Chart QA benchmarks in terms of chart distribution (Figure 5) and, most importantly, model performance. In particular, our analysis focuses on three representative real-image-based benchmarks: ChartQA [23], CharXiv [40], and ChartQAPro [24].

We evaluate all models using the prompt specified in Figure 26. Due to the inference costs associated with large reasoning models, we limited the evaluation of o3 (high) and Gemini-2.5-Pro only on our benchmark. Models and their configurations can be found in Section 4 and Appendix D, respectively.

For each benchmark, we report the average performance on their respective test sets. In the case of ChartQAPro, we exclude conversational and unanswerable questions to align the evaluation with our benchmark's scope. As shown in Figure 7, there is a large discrepancy among model performance in CHARTMUSEUM, which does not occur in other datasets.

Note that we do not include other specialized chart understanding models, such as TinyChart-3B [44] and ChartGemma-3B [25] in our evaluation, since they perform significantly worse than non-SOTA open-source LVLMs according to [24].

## A.2    Examples

We show representative examples of *(image, question, answer)* tuples from CHARTMUSEUM over diverse question and reasoning categories. An overall reference to the examples can be found in Table 7.

# B    Synthetic Dataset Details

## B.1    Synthetic Visual Reasoning Case Study

### B.1.1    Dataset Creation

We present 5 different chart types and 13 different visual reasoning questions in Table 8. Density and Histogram questions test the model's ability to categorize each figure by shape, while Line, Scatter, and Violin questions test the model's ability to compare each figure's characteristic such as cluster density or height.

| Examples | Figure # |
|---|---|
| *Question Categories* | |
| Textual Reasoning Questions | Figure 8, 12 |
| Visual/textual reasoning Questions | Figure 9, 13 |
| Synthesis Reasoning Questions | Figure 10, 14 |
| Visual Reasoning Questions | Figure 11, 15 |
| *Visual Task Taxonomy* | |
| Symbol Selection | Figure 16 |
| Visual Comparison | Figure 17 |
| Trajectory Tracking | Figure 18 |
| X/Y Value Identification | Figure 19 |
| *Others* | |
| Strategy Errors | Figure 20, 21, 22 |

Table 7: References to all example figures from CHARTMUSEUM in different categories.

| Chart Type | Questions |
|---|---|
| Density | Which figure is uni/bi/trimodal? |
| Histogram | Which figure is right/left-skewed, uniformly distributed, or symmetric (bell-shaped)? |
| Line | Which figure shows the most volatile or stable pattern? |
| Scatter | Which figure shows the strongest or weakest clustering pattern? |
| Violin | Which figure has the largest or smallest range? |

Table 8: Chart type and questions for our visual reasoning synthetic dataset used in Section 2.3

For each chart type, we write a python script that randomly selects the figure index with certain property such as left-skewed for histogram and generates the rest of the figures accordingly. We add small randomness to each figure to ensure that all images appear different from one another. Example images can be found in Figure 25.

### B.1.2 Human Evaluation and Results

We sample 20 examples for each chart type where $n \in [3, 6, 9]$, except for Histogram, where $n \in [4, 6, 9]$. We recruit two computer science researchers who were not involved in the synthetic dataset creation process. Each annotator is assigned 10 examples for each chart type. Additionally, we randomize the order of the questions within each figure to prevent memorization. The annotators provide corresponding alphabet letter and we use exact-match to evaluate the answer. In total, human annotators have annotated 1,560 image-question pairs.

We compare our human evaluation result and model performance in Figure 25. Human performance for both *Overlay* and *Subplot* setup is consistently higher or similar to model across different chart types. While human accuracy generally remains perfect as $n$ increases, the model accuracy tends to decrease or plateau at similar or lower accuracy than human.

## C    Task Taxonomy Examples and Details

### C.1    Task Taxonomy Examples

We present examples of the four different visual task categories in figures below:

- Visual Comparison: Figure 17

- Symbol Selection: Figure 16

- Trajectory Tracking and Judgment: Figure 18

| Task Type | Count |
|---|---|
| Size | 13 |
| Height | 5 |
| Color Scale | 3 |
| Variance | 3 |
| Spatial Distance | 2 |
| Pattern Comparison | 1 |
| Any Visual Comparison | 26 |

Table 9: Number of randomly sampled ($N = 50$) ChartMuseum (excluding *textual reasoning*) questions that involve each type of visual comparison. A question can involve multiple types of visual comparison.

- X/Y-Value Identification: Figure 19

A detailed visual comparison breakdown can be found in Table 9.

## C.2 Strategy Error Examples

Strategy Error examples can be found in Figure 20, Figure 21, and Figure 22. The synthesis example in Figure 14 is also an example of a strategy error for Claude-3.7-Sonnet and Gemini-2.5-Pro.

## D Model Details

We use the official APIs for proprietary LVLMs. The checkpoints we used across all experiments in this work can be found in Table 10. For open-source models, we run the model inference using NVIDIA RTX A6000 GPUs.

During inference, we use a temperature of 1, which is recommended for reasoning models, for o3 (high), o4-mini (high), Claude-3.7-Sonnet Extended-Thinking, and Gemini-2.5-Pro. For the remaining models, we use a temperature of 0. We leave all other hyperparameters their default values.

| Model | Checkpoint |
|---|---|
| GPT-4o | gpt-4o-2024-11-20 |
| GPT-4.1-mini | gpt-4.1-mini-2025-04-14 |
| GPT-4.1 | gpt-4.1-2025-04-14 |
| o3 (high) | o3-2025-04-16 |
| o4-mini (high) | o4-mini-2025-04-16 |
| Claude-3.5-Sonnet | claude-3-5-sonnet-20241022 |
| Claude-3.7-Sonnet | claude-3-7-sonnet-20250219 |
| Gemini-1.5-Flash | gemini-1.5-flash |
| Gemini-1.5-Pro | gemini-1.5-pro |
| Gemini-2.5-Pro | gemini-2.5-pro-preview-03-25 |

Table 10: LVLM checkpoints

The inference time for open-source models on the benchmark varies between 5 to 20 minutes per model, depending on size. For proprietary LVLMs, costs varies: o3 (high) runs at \$100; Gemini-2.5-Pro runs at \$50, Claude-3.5-Sonnet, Claude-3.7-Sonnet, and o4-mini (high) each cost \$30. GPT-4o and GPT-4.1 are around \$10 and the remaining models are within \$1 each.

We use gpt-4.1-mini-2025-04-14 to evaluate the answer correctness of all experiments with the prompt in Figure 27. The cost for evaluating a model's performance on our benchmark is \$0.2.

| CHARTMUSEUM | Overall (162) |
|---|---|
| *Open-Source Models (2B and 3B)* | |
| InternVL3-2B | 14.8 |
| Qwen2.5-VL-3B | 14.8 |
| *Open-Source/Specialized Models (7B and 8B)* | |
| Qwen2.5-VL-7B | 25.3 |
| InternVL3-8B | 28.4 |
| Bespoke-MiniChart-7B | 37.7 |
| *Open-Source Models (32B and more)* | |
| InternVL3-38B | 33.3 |
| InternVL3-78B | 34.0 |
| Qwen2.5-VL-32B | 37.0 |
| Pixtral-Large-124B | 36.4 |
| Qwen2.5-VL-72B | 40.1 |
| *Proprietary Models* | |
| Gemini-1.5-Flash | 33.3 |
| Gemini-1.5-Pro | 41.4 |
| GPT-4o | 40.1 |
| GPT-4.1 | 51.9 |
| GPT-4.1-mini | 54.9 |
| Claude-3.5-Sonnet | 53.7 |
| Claude-3.7-Sonnet | 63.0 |
| *Proprietary Models - Reasoning* | |
| o3 (high) | 67.3 |
| o4-mini (high) | 64.8 |
| Claude-3.7-Sonnet (think) | 64.8 |
| Gemini-2.5-Pro | 65.4 |

Table 11: Accuracy performance comparison of models on the development set of the CHARTMU-SEUM benchmark.

**Example: *Textual Reasoning* Question**

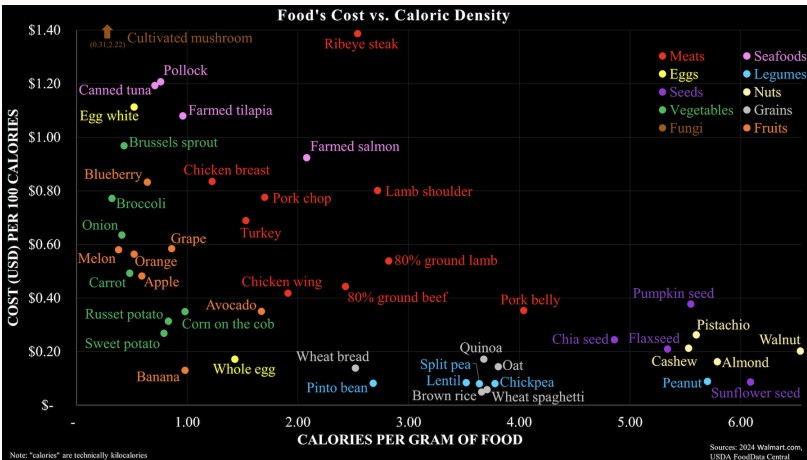

**Question:** Suppose Bob has $5 and he want to eat a type of Meats that can maximize the weight (in grams), which meat in the Meats category should Bob buy?

**Answer:** Chicken wing

---

**Qwen2.5-VL 72B-Instruct**

<think>
To determine which meat Bob should buy to maximize the grams of food with $5, we need to consider both the cost per 100 calories and the calories per gram of food. The goal is to find the meat that provides the most grams of food for the least cost.

1. **Identify the meats in the chart**: The meats listed are Ribeye steak, Farmed salmon, Chicken breast, Pork chop, Lamb shoulder, 80% ground lamb, Chicken wing, 80% ground beef, and Pork belly.

2. **Evaluate the cost per 100 calories**: We need to find the meat with the lowest cost per 100 calories, as this will allow Bob to buy more calories (and thus more grams of food) with his $5.

3. **Compare the costs**:
   . . . omitted . . .
   - Pork belly $\sim$ $0.40 per 100 calories

4. **Determine the most cost-effective meat** The meats with the lowest cost per 100 calories are Chicken wing, 80% ground beef, and Pork belly, all at $\sim$$0.40 per 100 calories.

5. **Consider calories per gram** Among these, we need to check which one has the highest calories per gram to maximize the grams of food.

6. **Conclusion**: Pork belly has the highest calories per gram at 4.0, making it the most efficient in terms of grams of food per dollar.

</think>
<answer>
Pork belly ✗
</answer>

Figure 8: An example from CHARTMUSEUM where textual reasoning is required. We include a human solution at the bottom (not included in the benchmark).

**Gemini-2.5-Pro**

<think>
To maximize the weight (in grams) of meat purchased for $5, Bob needs to find the meat with the lowest cost per gram. The cost per gram can be calculated from the chart's axes:

Cost per gram = (Cost per 100 Calories / 100) * (Calories per Gram) Cost per gram = (Y-axis value / 100) * (X-axis value)

Maximizing the weight purchased for a fixed amount of money is equivalent to minimizing the cost per gram. Therefore, we need to find the meat for which the product of its X-coordinate (Calories per Gram) and its Y-coordinate (Cost per 100 Calories) is the minimum.

Let's estimate the coordinates (X, Y) and calculate the product (X * Y) for each meat (red dots):

...calculation for some meats omitted here...
* Turkey: (approx. 1.9, $0.68) => Product ≈ 1.9 * 0.68 ≈ 1.29
* Chicken wing: (approx. 1.8, $0.48) => Product ≈ 1.8 * 0.48 ≈ **0.86**
* 80% ground beef: (approx. 2.5, $0.42) => Product ≈ 2.5 * 0.42 ≈ 1.05
* Pork belly: (approx. 4.1, $0.40) => Product ≈ 4.1 * 0.40 ≈ 1.64

Comparing the products, Chicken wing has the smallest product (approximately 0.86). A smaller product means a lower cost per gram, and therefore, more grams per dollar. Thus, Bob should buy Chicken wings to maximize the weight of meat for his $5.
</think>
<answer>
Chicken Wing ✓
</answer>

**o4-mini (high)**

<think>
1. To maximize grams of meat with a fixed budget ($5), Bob should choose the meat with the lowest cost per gram.

2. From the chart, "Meats" are the red points. For each, estimate:
- Cost per 100 cal (y-axis)
- Calories per gram (x-axis)

3. Compute cost per gram = (cost per 100 cal / 100 cal) × (calories per gram).

4. Approximate for each red point:
     1. Ribeye steak: $1.35/100 × 2.5 ≈ \$0.0338$ per g
     2. Chicken breast: $0.83/100 × 1.2 ≈ \$0.00996$ per g
     3. Turkey: $0.67/100 × 1.5 ≈ \$0.0101$ per g
      ...omitted ...
     7. 80% ground beef: $0.45/100 × 3.0 ≈ \$0.0135$ per g
     8. 80% ground lamb: $0.55/100 × 2.7 ≈ \$0.0149$ per g
     9. Pork belly: $0.35/100 × 3.7 ≈ \$0.0130$ per g

5. The lowest cost per gram among these is chicken wing (~$0.0082/g).

6. Therefore, with $5, Bob gets the most grams by buying chicken wings.
</think>
<answer>
Chicken wing ✓
</answer>

Figure 8 continued.

**Example:** *Visual/Textual Reasoning* **Question**

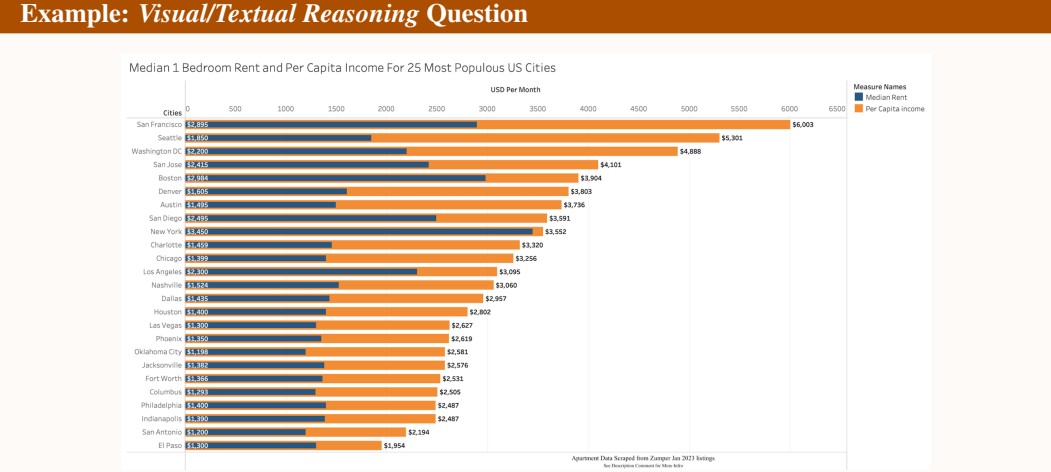

**Question:** Which US city has the highest fraction of income used for rent?
**Answer:** New York

**Claude-3.7-Sonnet (*Textual Reasoning Solution*)**

\<think\>
To find which US city has the highest fraction of income used for rent, I need to calculate the ratio of median rent to per capita income for each city in the chart.

For each city, I'll calculate: (Median Rent / Per Capita Income)
- . . . . . .
- Austin: $1,495 / $3,736 = 0.400
- San Diego: $2,495 / $3,591 = 0.695
- New York: $3,450 / $3,552 = 0.971
- Charlotte: $1,459 / $3,320 = 0.439
- Chicago: $1,399 / $3,256 = 0.430
- . . . . . .

Based on these calculations, New York has the highest fraction of income used for rent at 0.971 (or about 97.1% of per capita income).
\</think\>
\<answer\>
New York ✓
\</answer\>

**Human (*Visual Reasoning* Solution)**

The income is represented by the orange color and the rent is represented by the blue color. To find the city where the highest fraction of income is used for the rent, we need to find the city where the orange bar is overlapped by the blue bar the most. It is clear that almost the entire bar for New York is in blue. Therefore the answer is New York.

There is no need to mathematically calculate and compare the exact fractions. ✓

Figure 9: An example from CHARTMUSEUM where either textual reasoning or visual reasoning is required. We include a human solution at the bottom (not included in the benchmark).

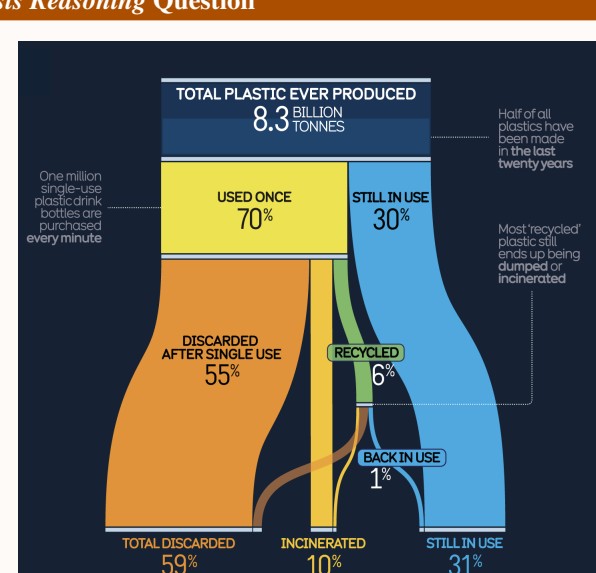

**Question:** How many tonnes of total plastic ever produced were recycled before being incinerated?

**Answer:** 0.083 billion tonnes

**Claude-3.7-Sonnet**

<think>
. . . Previous CoTs omitted . . .

There's no direct data in the chart that tells me exactly how much recycled plastic was incinerated. The chart shows that 10% of total plastic (0.8 billion tonnes) was incinerated, but it doesn't break down how much of that was previously recycled.

. . .
</think>
<answer>
The chart does not provide specific data on how many tonnes of recycled plastic were later incinerated. ✗
</answer>

Figure 10: An example from CHARTMUSEUM where both visual reasoning and textual reasoning are required. We include a human solution at the bottom (not included in the benchmark). Key reasoning steps is highlighted in green.

**o3 (high)**

. . . Reasoning block omitted . . .
<think>
The Sankey diagram shows the fates of the 8.3 billion tonnes of plastic ever produced.
- 6 % of that total (0.498 billion tonnes) entered a "Recycled" stream.
- Of the recycled stream, 1 % of the global total (0.083 billion tonnes) made it "Back in use."
- The remaining part of the recycled stream, 5 % of the global total, went on to the "Incinerated" box.

Therefore, the plastic that was first recycled and then incinerated amounts to 5 % of 8.3 billion tonnes. Calculation: 8.3 billion tonnes × 0.05 = 0.415 billion tonnes (≈ 0.4 billion tonnes).
</think>
<answer>
About 0.4 billion tonnes. ✗
</answer>

**Gemini-2.5-Pro**

. . . Reasoning block omitted . . .
<think>
. . . The infographic shows that the stream of recycled plastic (6%) splits, with 1% going back into use and the remaining 5% contributing to the final "Total Discarded" (59%) and "Incinerated" (10%) pools. However, the infographic does not specify how that 5% (0.415 billion tonnes) of recycled-but-not-reused plastic is divided between being discarded and being incinerated. It only gives the total amount incinerated (10% or 0.8 billion tonnes) from all sources. . . .
</think>
<answer>
We cannot determine the exact amount of plastic that followed the path: Recycled -> Incinerated. We only know the total amount incinerated is 0.8 billion tonnes and the total amount recycled is about 0.5 billion tonnes. ✗
</answer>

**Human Solution**

Based on the flow shown in the chart, we can see that 6% of the total plastic (8.3 Billion Tonnes) was sent for recycling and 1% of the total plastic went "Back in Use" after recycling. This leaves 6% - 1% = 5% of the total plastic that was recycled but did not end up back in use.

On the left side, we see that the "Total Discarded" is 59%, while "Discarded After Single Use" is 55%. Therefore, the difference (59% - 55% = 4%) represents the recycled plastic that ended up being discarded.

In conclusion, there are 6% - 1% - 4% = 1% of total plastic that were recycled before being incinerated, which is 0.01 * 8.3 = 0.083 Billion Tonnes. ✓

Figure 10 continued.

**Example: *Visual Reasoning* Question**

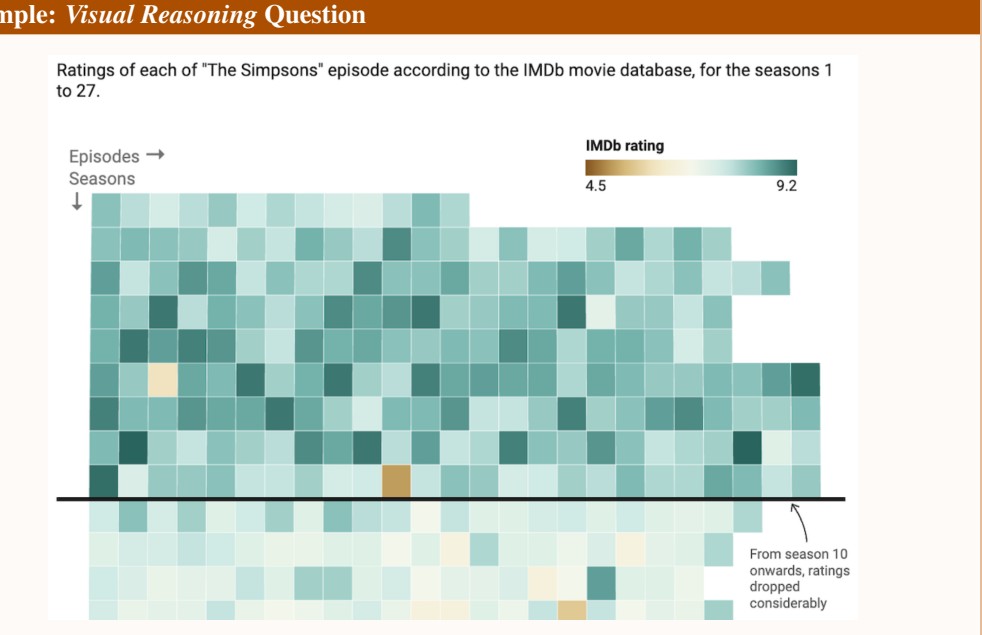

Ratings of each of "The Simpsons" episode according to the IMDb movie database, for the seasons 1 to 27.

**Question:** Based on the ratings for "The Simpsons" across seasons and episodes in the chart, what is the median number of episodes per season from Season 1 through Season 9? Note that Season 9 has 25 episodes.

**Answer:** 24

---

**Qwen2.5-VL 72B-Instruct**

<think>
To find the median number of episodes per season from Season 1 through Season 9, I need to:
1. Determine the number of episodes in each season from 1 to 9
2. Arrange these numbers in ascending order
3. Find the middle value (median)

From the image, I can see a heatmap of episode ratings, but it's difficult to count the exact number of episodes per season just by looking at the colored squares. The question states that Season 9 has 25 episodes, but doesn't provide the episode counts for the other seasons.

Without being able to clearly count the number of episodes per season from the image, I cannot determine the exact episode counts for Seasons 1-8. Therefore, I don't have enough information from the image alone to calculate the median.
</think>
<answer>
I cannot determine the median number of episodes per season from the image provided. ✗
</answer>

Figure 11: An example from CHARTMUSEUM where visual reasoning is required. Key reasoning step from the model is highlighted in green. We also include a human solution at the bottom (not included in the benchmark).

**GPT-4.1**

<think>
To find the median number of episodes per season from Season 1 through Season 9, I need to count the number of episode squares in each of the first 9 rows (each row represents a season, and each square an episode).I will then list these counts, sort them, and find the median value (the middle value in the sorted list).

Counting the episodes in each of the first 9 seasons:
- Season 1: 13 episodes
- Season 2: 22 episodes
- Season 3: 24 episodes
- Season 4: 22 episodes
- Season 5: 22 episodes
- Season 6: 25 episodes
- Season 7: 25 episodes
- Season 8: 25 episodes
- Season 9: 25 episodes

Now, I will list these numbers in order: 13, 22, 22, 22, 24, 25, 25, 25, 25. Since there are 9 seasons, the median is the 5th value in the sorted list.
</think>
<answer>
24 ✓
</answer>

**Mistral-Large-124B**

<think>
To find the median number of episodes per season from Season 1 through Season 9, we need to list the number of episodes for each season and then find the median value.
From the chart, we can see the following number of episodes per season:
- Season 1: 13 episodes
- Season 2: 22 episodes
- Season 3: 24 episodes
- Season 4: 22 episodes
- Season 5: 22 episodes
- Season 6: 25 episodes
- Season 7: 25 episodes
- Season 8: 25 episodes
- Season 9: 25 episodes

Now, we list these numbers in ascending order: 13, 22, 22, 22, 24, 25, 25, 25, 25
Since there are 9 seasons, the median will be the 5th value in this ordered list.
</think>
<answer>
24 ✓
</answer>

**Human Solution**

We know that Season 9 has 25 episodes. From the chart, we can see that the last four Seasons (6, 7, 8, 9) have the same number of grids (also the longest), which implies that all 4 episodes have 25 episodes. This finding suggests that, to find the median (5th), we only need to find the second-longest row in the chart. We observe that there is a row with one fewer grid on the right, meaning it has 24 episodes. That would be the answer.

We don't need to count episodes from each Season to answer the question. ✓

Figure 11 continued.

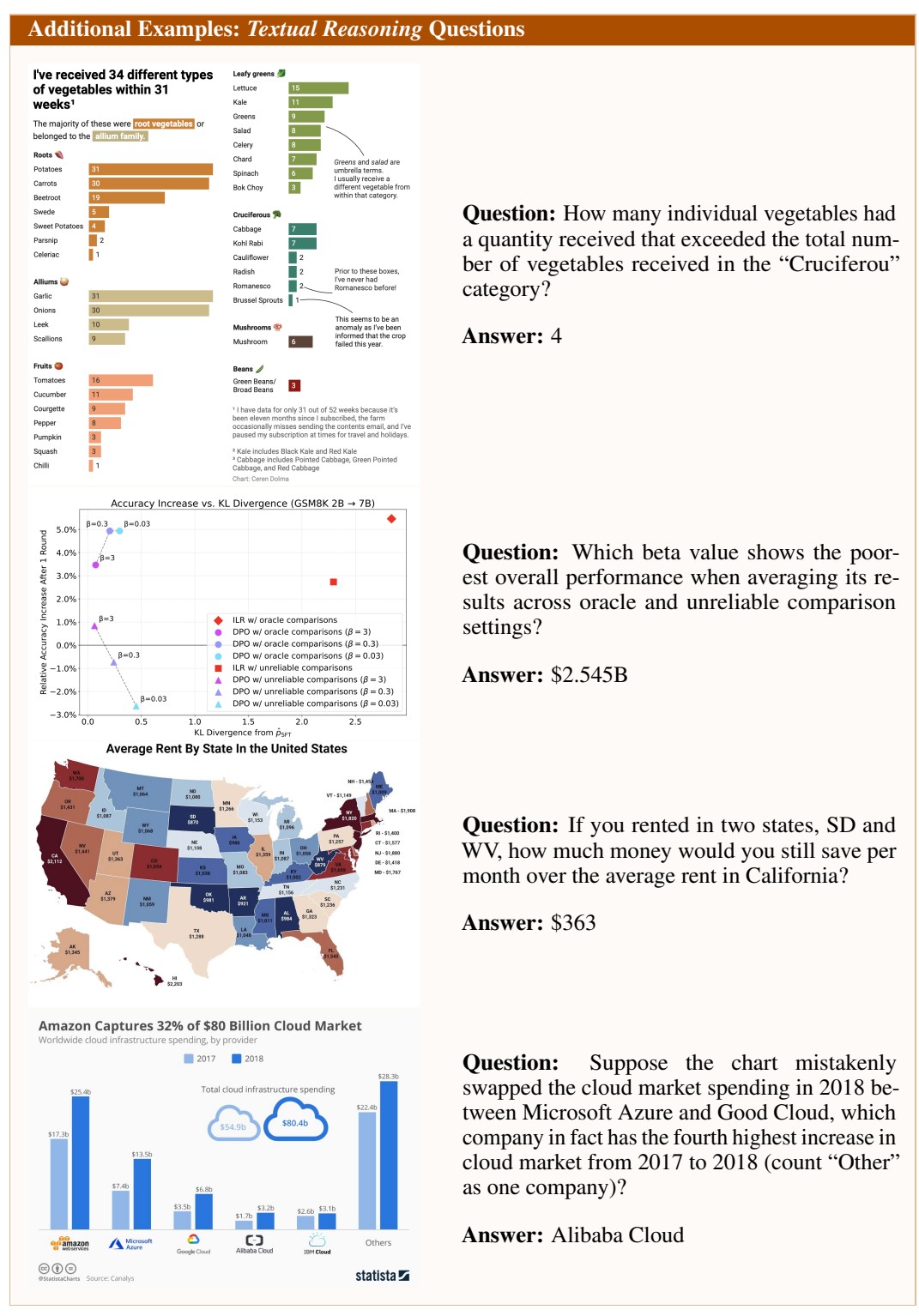

**Question:** How many individual vegetables had a quantity received that exceeded the total number of vegetables received in the "Cruciferou" category?

**Answer:** 4

**Question:** Which beta value shows the poorest overall performance when averaging its results across oracle and unreliable comparison settings?

**Answer:** $2.545B

**Question:** If you rented in two states, SD and WV, how much money would you still save per month over the average rent in California?

**Answer:** $363

**Question:** Suppose the chart mistakenly swapped the cloud market spending in 2018 between Microsoft Azure and Good Cloud, which company in fact has the fourth highest increase in cloud market from 2017 to 2018 (count "Other" as one company)?

**Answer:** Alibaba Cloud

Figure 12: Additional examples from CHARTMUSEUM where textual reasoning is required.

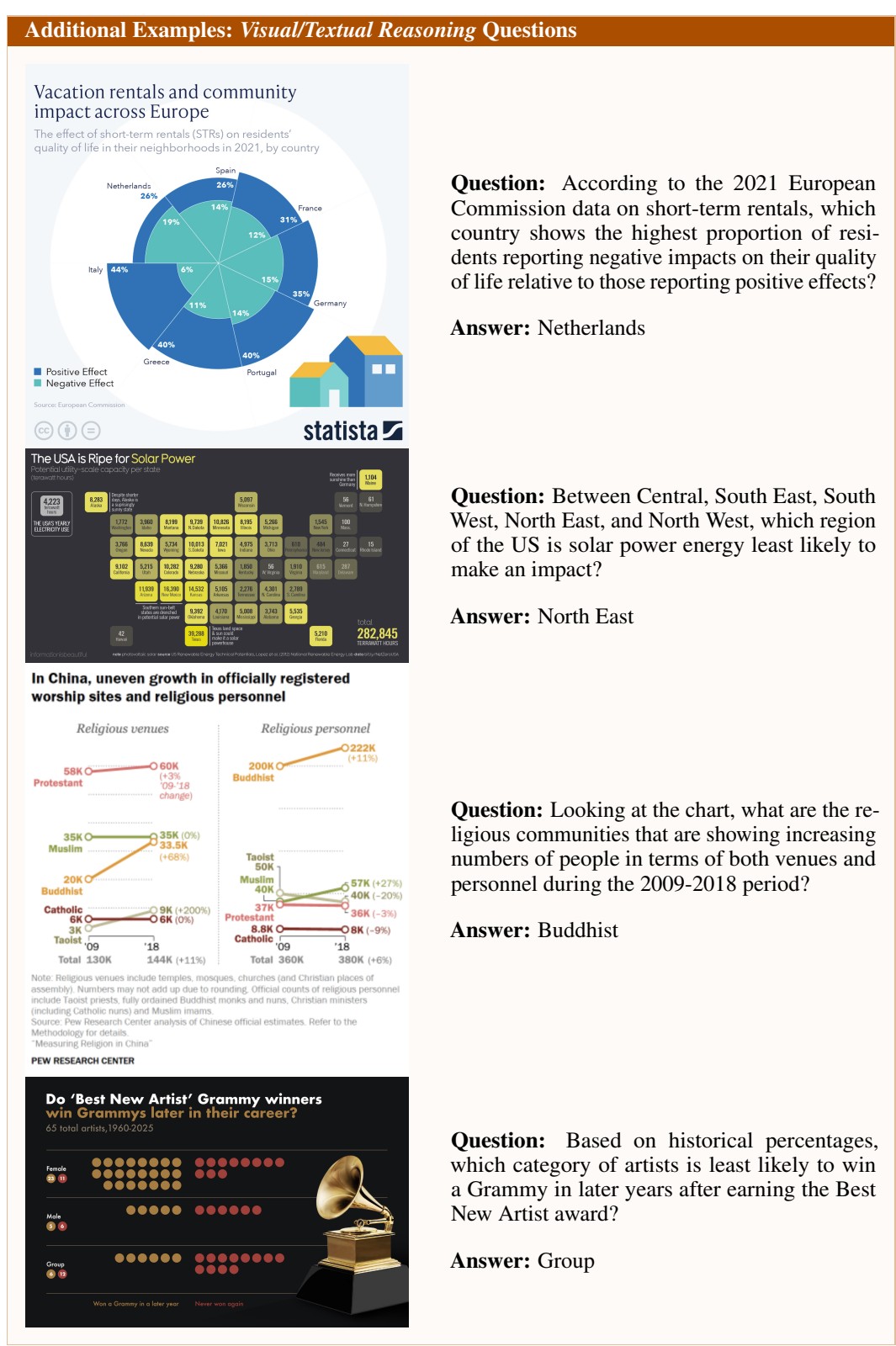

**Question:** According to the 2021 European Commission data on short-term rentals, which country shows the highest proportion of residents reporting negative impacts on their quality of life relative to those reporting positive effects?

**Answer:** Netherlands

**Question:** Between Central, South East, South West, North East, and North West, which region of the US is solar power energy least likely to make an impact?

**Answer:** North East

**Question:** Looking at the chart, what are the religious communities that are showing increasing numbers of people in terms of both venues and personnel during the 2009-2018 period?

**Answer:** Buddhist

**Question:** Based on historical percentages, which category of artists is least likely to win a Grammy in later years after earning the Best New Artist award?

**Answer:** Group

Figure 13: Additional examples from CHARTMUSEUM where either textual reasoning or visual reasoning is required.

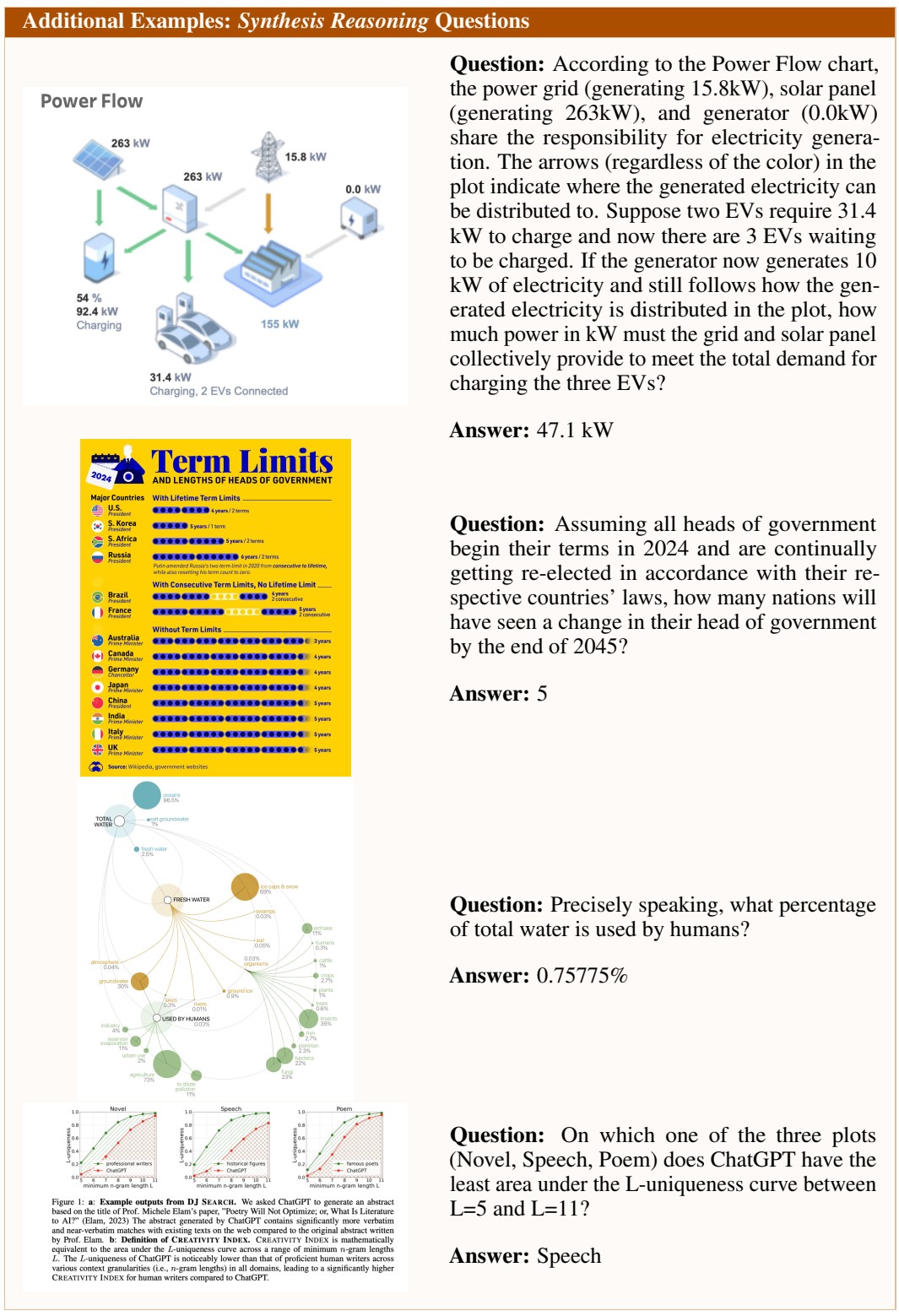

**Question:** According to the Power Flow chart, the power grid (generating 15.8kW), solar panel (generating 263kW), and generator (0.0kW) share the responsibility for electricity generation. The arrows (regardless of the color) in the plot indicate where the generated electricity can be distributed to. Suppose two EVs require 31.4 kW to charge and now there are 3 EVs waiting to be charged. If the generator now generates 10 kW of electricity and still follows how the generated electricity is distributed in the plot, how much power in kW must the grid and solar panel collectively provide to meet the total demand for charging the three EVs?

**Answer:** 47.1 kW

**Question:** Assuming all heads of government begin their terms in 2024 and are continually getting re-elected in accordance with their respective countries' laws, how many nations will have seen a change in their head of government by the end of 2045?

**Answer:** 5

**Question:** Precisely speaking, what percentage of total water is used by humans?

**Answer:** 0.75775%

**Question:** On which one of the three plots (Novel, Speech, Poem) does ChatGPT have the least area under the L-uniqueness curve between L=5 and L=11?

**Answer:** Speech

Figure 1: **a: Example outputs from DJ SEARCH.** We asked ChatGPT to generate an abstract based on the title of Prof. Michele Elam's paper, "Poetry Will Not Optimize; or, What Is Literature to AI?" (Elam, 2023) The abstract generated by ChatGPT contains significantly more verbatim and near-verbatim matches with existing texts on the web compared to the original abstract written by Prof. Elam. **b: Definition of CREATIVITY INDEX.** CREATIVITY INDEX is mathematically equivalent to the area under the *L*-uniqueness curve across a range of minimum *n*-gram lengths *L*. The *L*-uniqueness of ChatGPT is noticeably lower than that of proficient human writers across various context granularities (i.e., *n*-gram lengths) in all domains, leading to a significantly higher CREATIVITY INDEX for human writers compared to ChatGPT.

Figure 14: Additional examples from CHARTMUSEUM where both visual reasoning and textual reasoning are required.

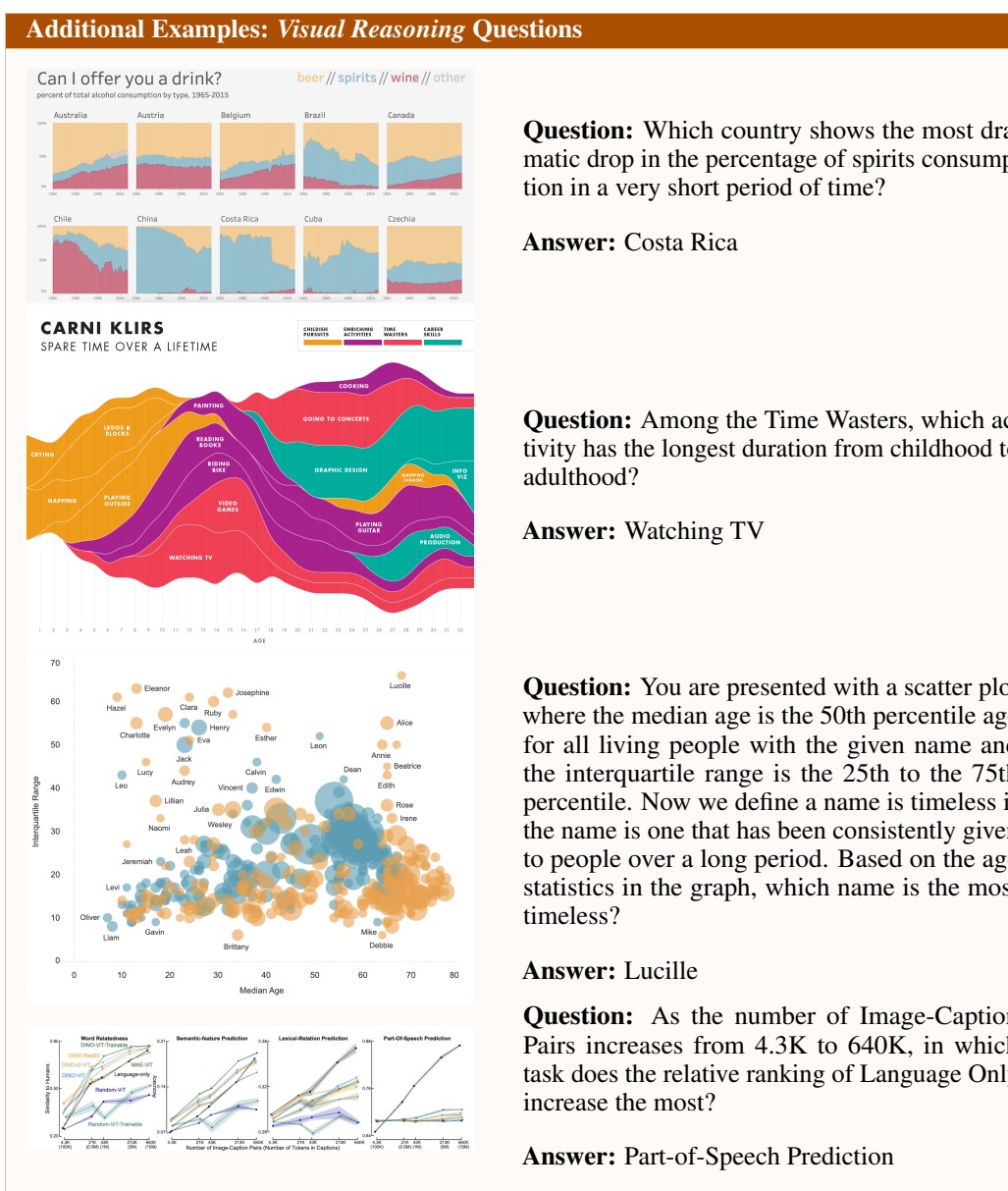

**Question:** Which country shows the most dramatic drop in the percentage of spirits consumption in a very short period of time?

**Answer:** Costa Rica

**Question:** Among the Time Wasters, which activity has the longest duration from childhood to adulthood?

**Answer:** Watching TV

**Question:** You are presented with a scatter plot where the median age is the 50th percentile age for all living people with the given name and the interquartile range is the 25th to the 75th percentile. Now we define a name is timeless if the name is one that has been consistently given to people over a long period. Based on the age statistics in the graph, which name is the most timeless?

**Answer:** Lucille

**Question:** As the number of Image-Caption Pairs increases from 4.3K to 640K, in which task does the relative ranking of Language Only increase the most?

**Answer:** Part-of-Speech Prediction

Figure 15: Additional examples from CHARTMUSEUM where visual reasoning is required.

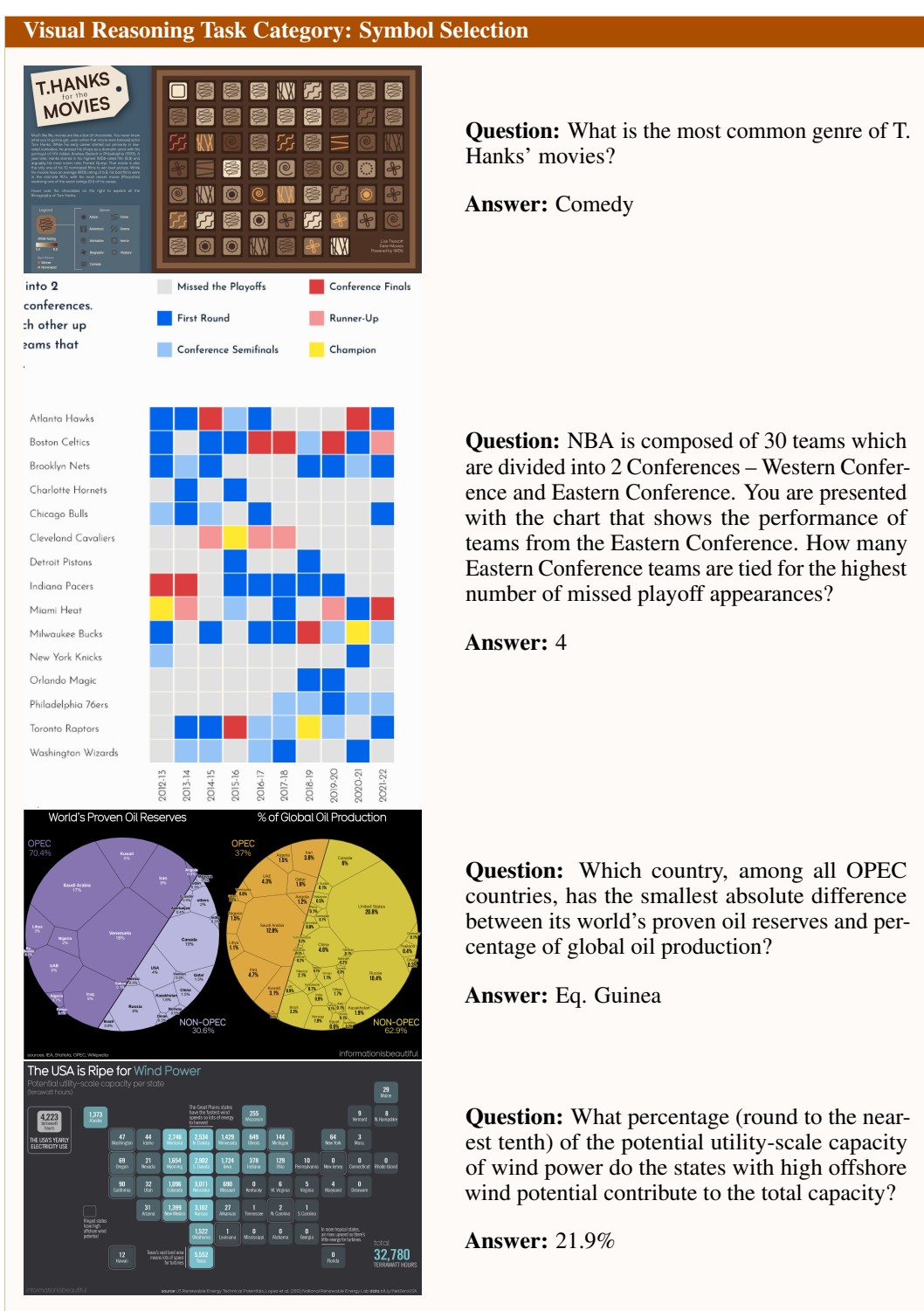

**Question:** What is the most common genre of T. Hanks' movies?

**Answer:** Comedy

**Question:** NBA is composed of 30 teams which are divided into 2 Conferences – Western Conference and Eastern Conference. You are presented with the chart that shows the performance of teams from the Eastern Conference. How many Eastern Conference teams are tied for the highest number of missed playoff appearances?

**Answer:** 4

**Question:** Which country, among all OPEC countries, has the smallest absolute difference between its world's proven oil reserves and percentage of global oil production?

**Answer:** Eq. Guinea

**Question:** What percentage (round to the nearest tenth) of the potential utility-scale capacity of wind power do the states with high offshore wind potential contribute to the total capacity?

**Answer:** 21.9%

Figure 16: Examples of ChartMuseum questions that involve a Symbol Selection task.

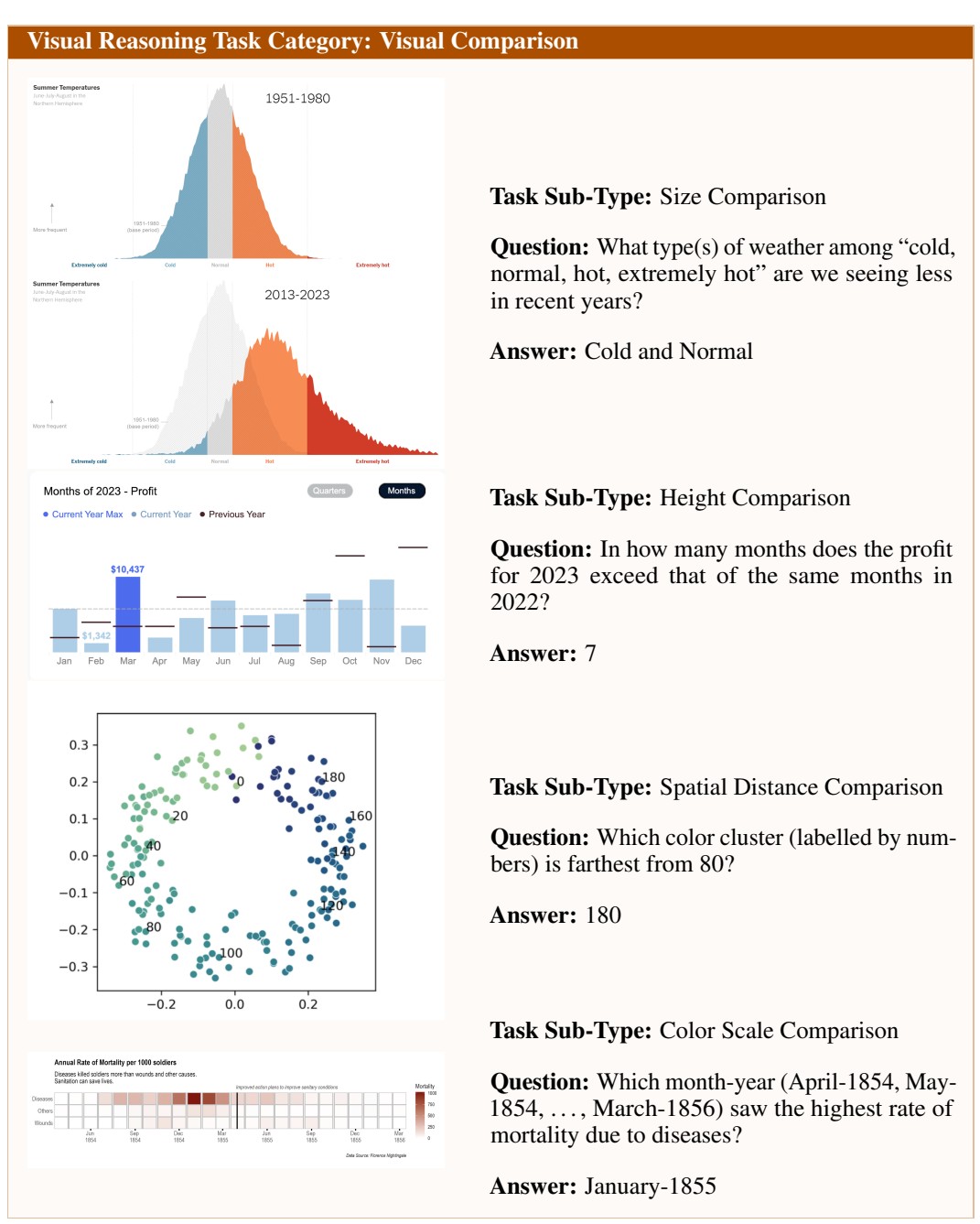

Figure 17: Examples of ChartMuseum questions that involve a Visual Comparison task.

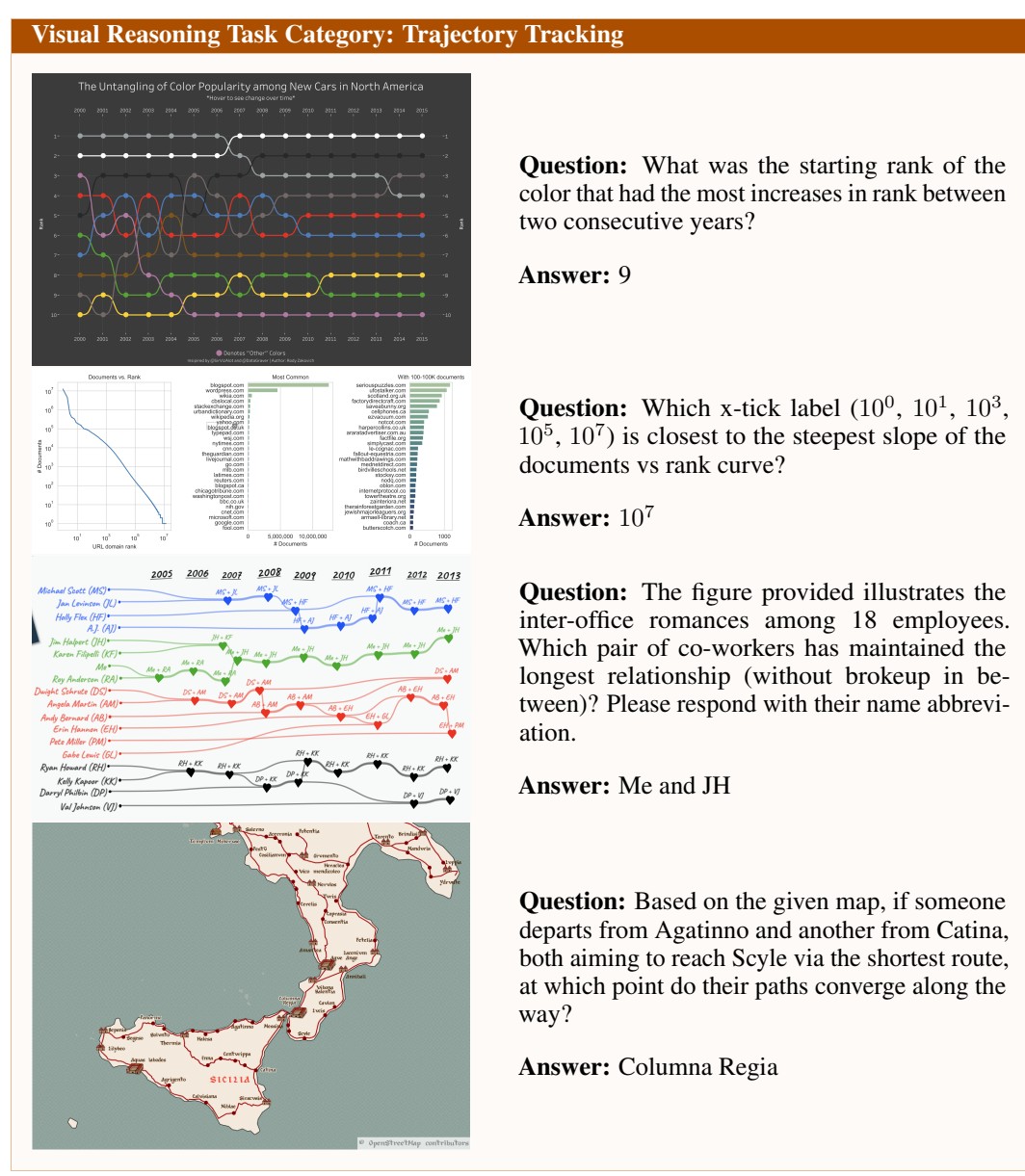

Figure 18: Examples of ChartMuseum questions that involve a Trajectory Tracking task.

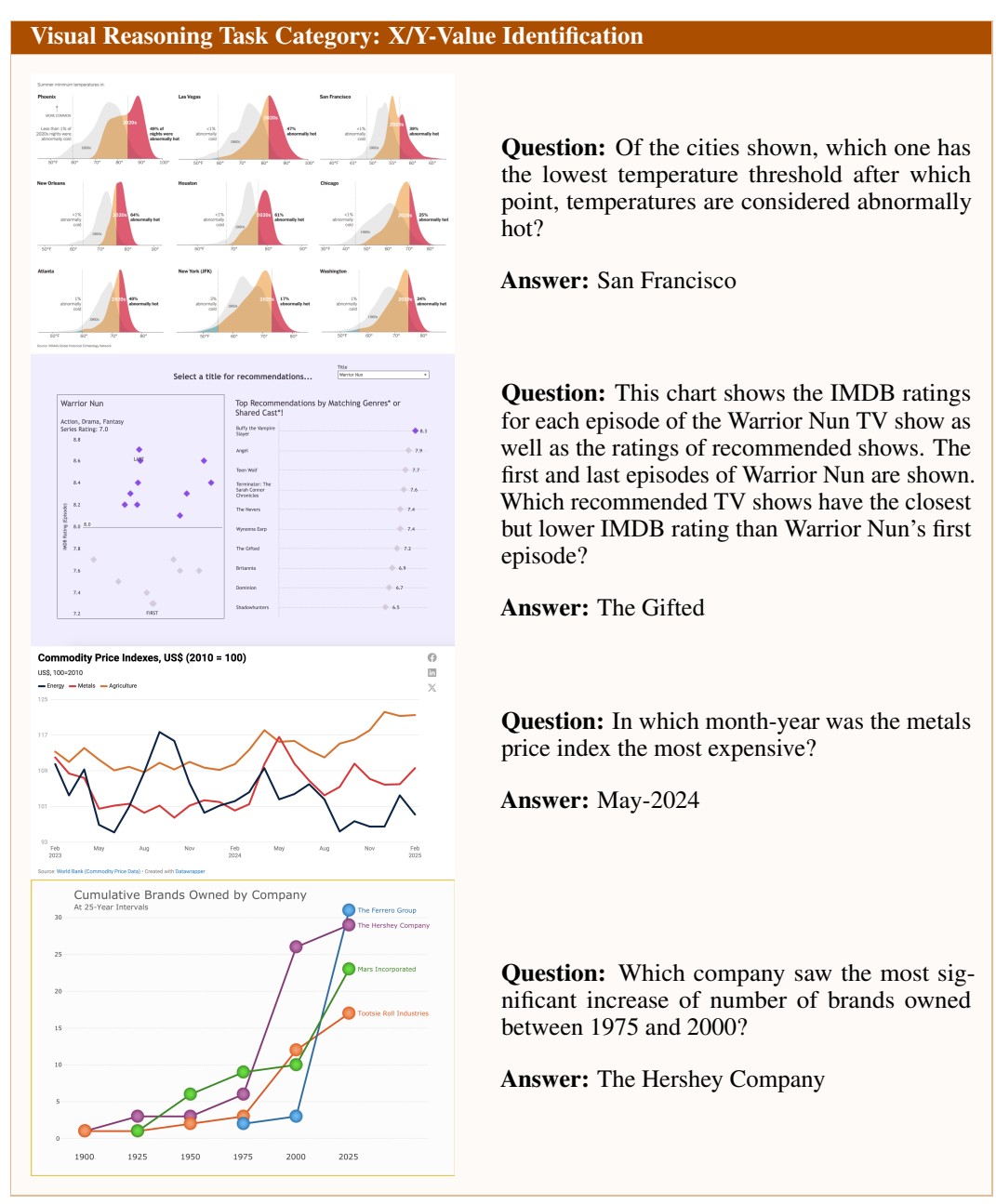

Visual Reasoning Task Category: X/Y-Value Identification

**Question:** Of the cities shown, which one has the lowest temperature threshold after which point, temperatures are considered abnormally hot?

**Answer:** San Francisco

**Question:** This chart shows the IMDB ratings for each episode of the Warrior Nun TV show as well as the ratings of recommended shows. The first and last episodes of Warrior Nun are shown. Which recommended TV shows have the closest but lower IMDB rating than Warrior Nun's first episode?

**Answer:** The Gifted

**Question:** In which month-year was the metals price index the most expensive?

**Answer:** May-2024

**Question:** Which company saw the most significant increase of number of brands owned between 1975 and 2000?

**Answer:** The Hershey Company

Figure 19: Examples of ChartMuseum questions that involve a X/Y-Value Identification task.

**Visual Reasoning Error Type: Strategy Error #1**

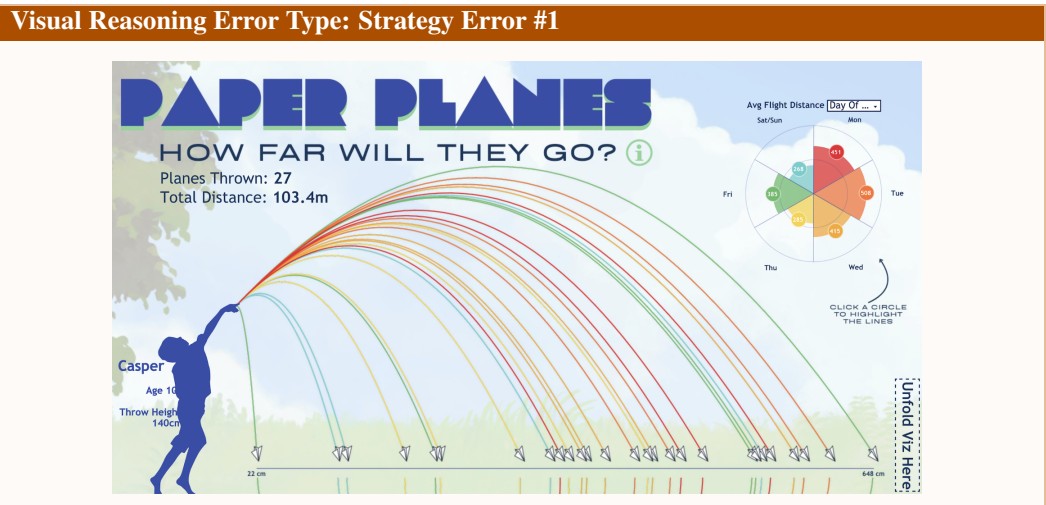

**Question:** During which day of the week does the paper plane show the highest range of flying distance?

**Answer:** Friday

---

**Claude-3.7-Sonnet**

<think>
Looking at the chart image, I can see a visualization of paper plane flights across different days of the week. On the right side of the image, there's a circular chart showing the average flight distances by day of the week.

. . .

The question asks for the day with the highest range of flying distance, not the highest average distance. However, the chart doesn't explicitly show the range (difference between maximum and minimum distances) for each day. It only shows average distances.

Based on the available information, Tuesday has the highest average distance at 508, but I don't have data on the range for each day. Without specific range information, I should note this limitation in my answer.
</think>
<answer>
Tuesday ✗
</answer>

Figure 20: Example of a Strategy Error.

<think>

. . .

8. The main visualization shows all the individual flights. To calculate the range for a specific day (e.g., Tuesday), I would need to identify all the orange lines, find the maximum distance achieved by an orange line, find the minimum distance achieved by an orange line, and subtract the minimum from the maximum.

9. Looking at the main visualization, while I can see lines of different colors reaching different distances, the exact minimum and maximum distance for *each specific color group (day)* is not explicitly labeled or easily measurable from the static image. I can see the overall minimum (22 cm) and overall maximum (648 cm), but I don't know which days these specific throws occurred on, nor do I know the specific min/max for *each* day.

10. For example, Tuesday (orange) has the highest average (508cm). Visually, some orange lines go very far (near the 648 cm mark), but it's impossible to tell the distance of the *shortest* orange line accurately from the image to calculate the range for Tuesday. The same applies to all other days.

11. Since the image provides average distances per day and overall min/max distances, but *not* the min/max distances *per day*, I cannot calculate the range for each day.

12. Therefore, based *only* on the information presented in the static image, it is impossible to determine which day has the highest range of flying distances. The necessary data (min/max distance for each day) is missing.
</think>
<answer>
Impossible to determine ✗
</answer>

**Error Explanation**

The range of each day must be inferred by looking at the flight arcs with that day's color, and comparing the difference between the shortest arc and the longest arc.

Claude-3.7-Sonnet is able to recognize that there is a main "visualization of paper plane flights across different days of the week", but does not realize that this can be used to answer the question. It instead focuses on the circular chart in the top right corner showing the average flight lengths by day.

Gemini-2.5-Pro recognizes that the flight arcs are important to answering the question, but gives up because value estimation is too difficult, not recognizing that visual range comparisons can still be made.

This question can be easily answered once realizing that the both the closest and the furthest lines are in green color, which is Friday. There is no need to measure the range of flying distance for other days.

Figure 20 continued.

From: 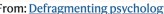 Defragmenting psychology

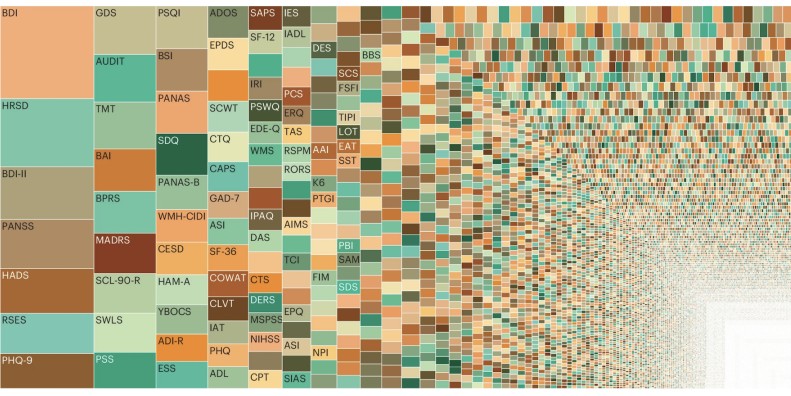

Each tile represents a measure (details of data in ref. [1]). Tile sizes are proportional to how often measures have been used. The mass of small tiles indicates a high level of fragmentation; thousands of measures are hardly ever reused. Larger tiles indicate measures that dominate the field in terms of usage; even for these, there is fragmentation, with multiple measures for the same construct (for example, BDI (Beck depression inventory), HADS (hospital anxiety and depression scale), PHQ-9 (patient health questionnaire-9) and more, which measure depression). Additional measure names can be found at this link (click on the first image to open another link, then hover over tiles to reveal).

**Question:** This chart shows the measures uses in psychology research papers as rectangles. Their areas are proportional to how often they are used. Which measure is the 10th most used measure?
**Answer:** TMT

**Claude-3.7-Sonnet**

<think>
I need to rank the measures by size (area) from largest to smallest, and then identify the 10th one.

Looking at the chart, I can see that the largest rectangles appear to be:
1. BDI (Beck Depression Inventory) - largest rectangle in the top left
2. HRSD (likely Hamilton Rating Scale for Depression) - large rectangle on left side
3. PANSS (Positive and Negative Syndrome Scale) - large brown rectangle
... omitted ...
7. GDS (Geriatric Depression Scale) - large light green rectangle
8. AUDIT (Alcohol Use Disorders Identification Test) - large green rectangle
9. BSI (Brief Symptom Inventory) - brown rectangle
10. PANAS (Positive and Negative Affect Schedule) - orange rectangle

So the 10th largest rectangle, representing the 10th most used measure, appears to be PANAS.

However, it's a bit difficult to precisely rank these by exact area from just visual inspection. Some rectangles like MADRS, BAI, and BDHI also appear to be relatively large, so I'm making my best estimate based on the visual sizes.
</think>
<answer>
PANAS (Positive and Negative Affect Schedule) ✗
</answer>

**Error Explanation**

While sizes of the 8th through 12th largest rectangles are incredibly similar, this question can be easily answered by recognizing that the boxes are ordered by size, top-to-bottom and left-to-right. Then it is clear that the 10th used measure is the 3rd box in column 2.

Figure 21: Example of a Strategy Error.

**Visual Reasoning Error Type: Strategy Error #3**

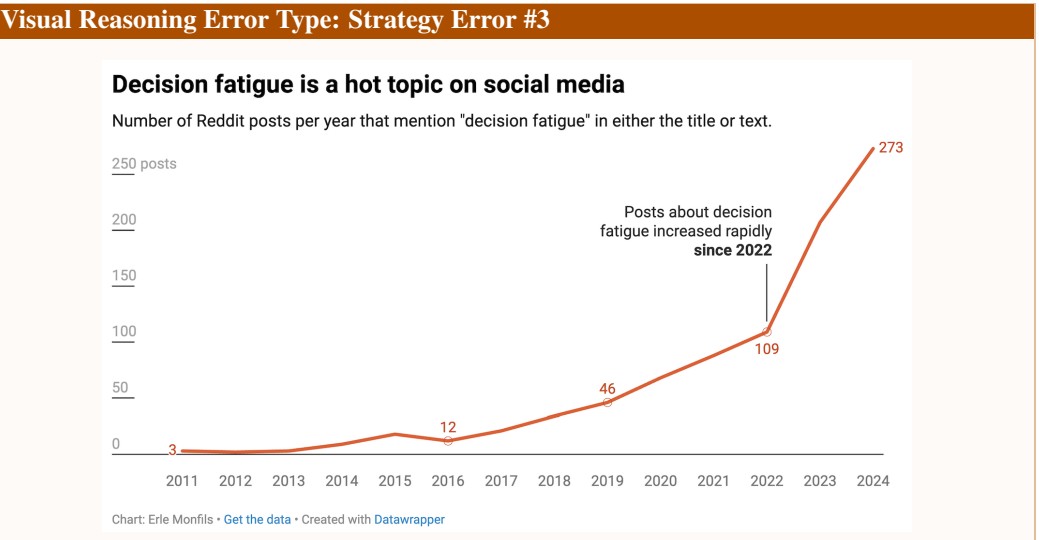

**Question:** During how many years was there 12 or fewer posts about decision fatigue?
**Answer:** 5

**Claude-3.7-Sonnet**

<think>
Looking at the chart, I need to identify the years where there were 12 or fewer posts about decision fatigue.
. . .
I can see specific data points:
- 2011: 3 posts
- 2016: 12 posts
- 2019: 46 posts
- 2022: 109 posts
- 2023: 273 posts
For the other years, I need to estimate based on the line graph:
- 2012-2015: The line stays very low, below 12 posts
- 2017-2018: The line is gradually rising but appears to be below 12 posts in 2017 and likely below 12 in 2018 as well
- 2020-2021: The line is between 46 and 109, so these years have more than 12 posts
. . .
Since I can't see the exact number for 2017 and it appears to be right around the threshold, I'll go with 6 years as the more conservative estimate.
</think>
<answer>
6 years  ✗
</answer>

**Error Explanation**

Since the y-axis is very low resolution, it is nearly impossible to determine the exact y-values for any unlabeled year. The intended solution is to compare the height of each point to the height of the year 2016. From this, it is clear that the value for year 2015 is higher than 12 (because the line slopes downwards to year 2016), and that the value for year 2017 is greater than 12 (because the line slopes upwards from year 2017 to year 2016).

Figure 22: Example of a Strategy Error.

Extract all text elements from the provided chart image (such as title,
annotated data, legend) and convert them into a structured JSON format.
Only extract data when the value is explicitly presented as a text element
on the chart.
If the value must be estimated (e.g., by reading from the axes), do not
extract it. Instead, write impossible to extract data in the value field.

Use the following JSON format:
{
    title: chart title,
    x_axis_label: x-axis label,
    y_axis_label: y-axis label,
    description: any text that is in outside of the chart,
    legend: [
    {label: category 1, color: color1},
    {label: category 2, color: color2},
    ... ]
    data: [
    {label: category 1, value: value 1, color: color1},
    {label: category 2, value: value 2, color: color2},
    ... ]
}
Do not include markdown formatting (e.g., ```json) or any comments.
Only return the raw JSON.

Figure 23: Text Extraction Prompt used for extracting only text information and color from charts. More details in Section 2.2.

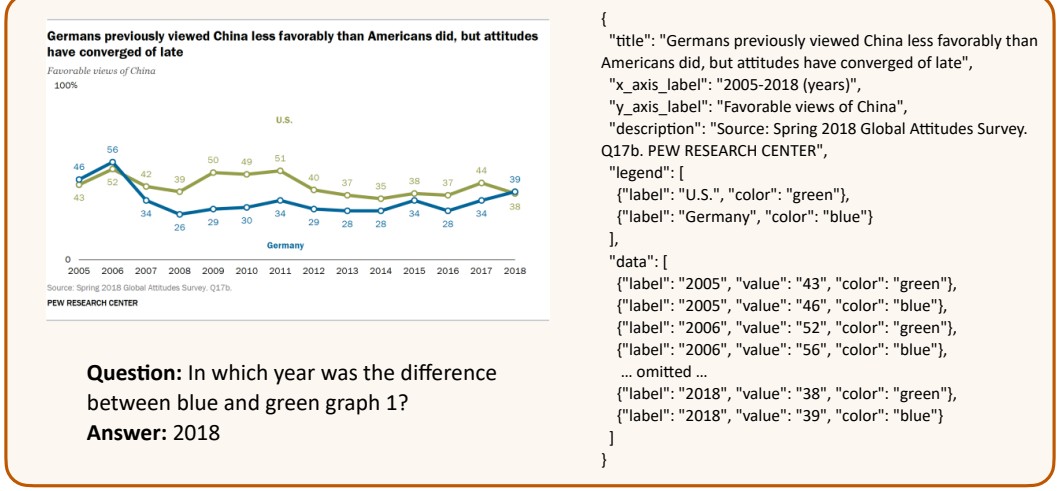

Figure 24: An example *(image, question, answer)* tuple from ChartQA [23]. The explicit information from the chart is extracted by Claude-3.7-Sonnet and is shown on the right. The extraction prompt is from Figure 23. We observe that most questions from ChartQA can be answered via extracted text element from the charts (Section 2.2).

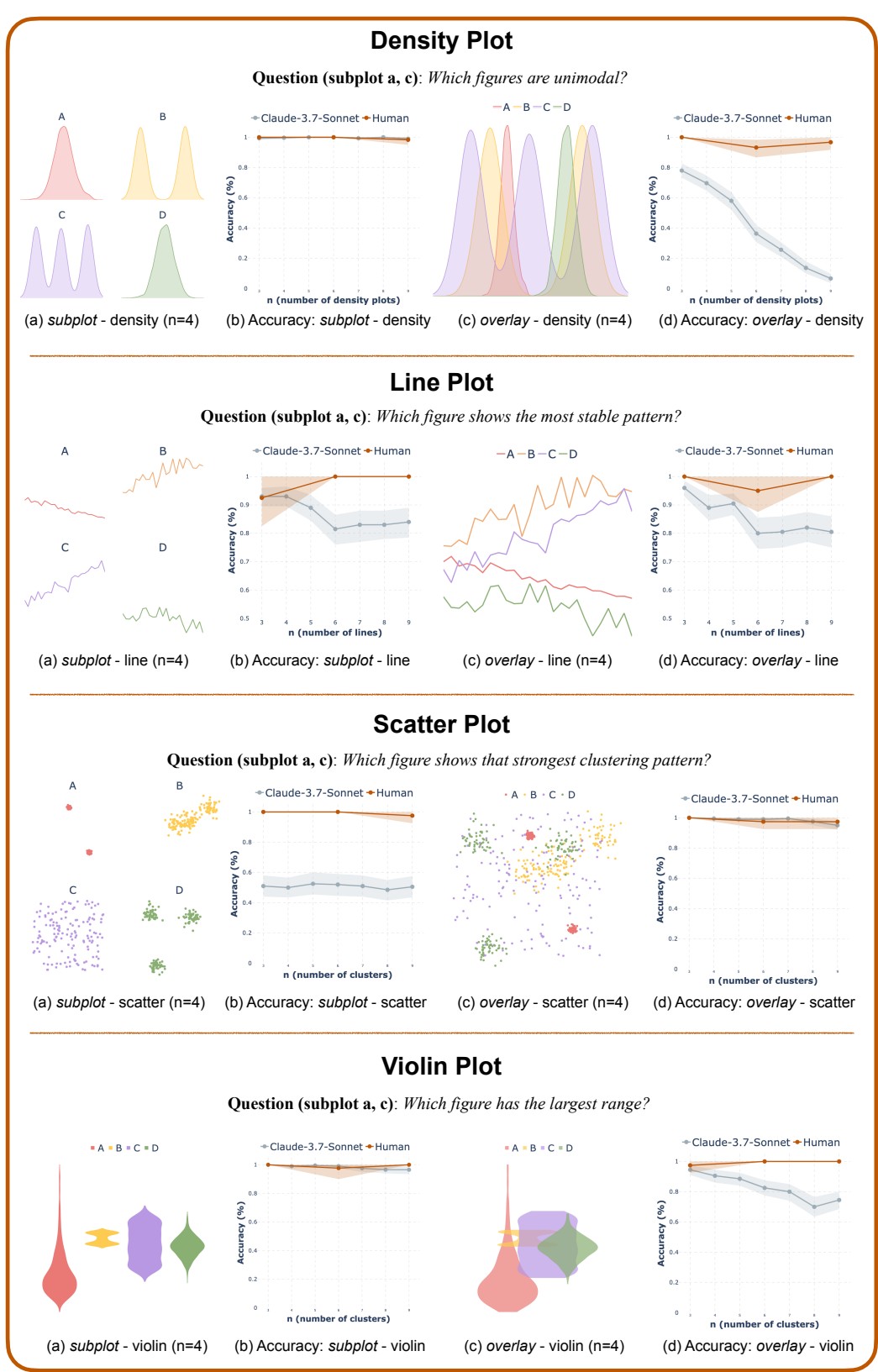

Figure 25: Visual reasoning case study over density/line/scatter/violin plots. We show the subplot and overlay setups on histograms in (a) and (c) with n=4. We compare Claude-3.7-Sonnet and human in answering the question over various values of $n \in [3, 9]$. Complete questions can be found in Table 8.

### Chain-of-Thought (CoT) Prompt for Chart QA

```
Please answer the question using the chart image.

Question: [QUESTION]

Please first generate your reasoning process and then provide the user with
the answer. Use the following format:

<think>
... your thinking process here ...
</think>
<answer>
... your final answer (entity(s) or number) ...
</answer>
```

Figure 26: Chain-of-Thought (CoT) Prompt for Chart QA

```
You are provided with a question and two answers. Please determine if these
answers are equivalent. Follow these guidelines:

1. Numerical Comparison:
   For decimal numbers, consider them as equivalent if their relative
   difference is sufficiently small.

   For example, the following pairs are equivalent:
     - 32.35 and 32.34
     - 90.05 and 90.00
     - 83.3% and 83.2%
     - 31 and 31%
   The following pairs are not equivalent:
     - 32.35 and 35.25
     - 90.05 and 91.05
     - 83.3% and 45.2%

   Note that if the question asks for years or dates, please do the exact
   match with no error tolerance.

2. Unit Handling:
   If only one answer includes units (e.g. '$', '%', '-', etc.), ignore the
   units and compare only the numerical values.

   For example, the following pairs are equivalent:
     - 305 million and 305 million square meters
     - 0.75 and 0.75%
     - 0.6 and 60%
     - $80 and 80
   The following pairs are not equivalent:
     - 305 million and 200 million square meters
     - 0.75 and 0.90%

3. Text Comparison:
   - Ignore differences in capitalization
   - Treat mathematical expressions in different but equivalent forms as
     the same (e.g., "2+3" = "5")

Question: [QUESTION]
Answer 1: [ANSWER1]
Answer 2: [ANSWER2]

Please respond with:
- "Yes" if the answers are equivalent
- "No" if the answers are different
```

Figure 27: Answer Evaluation Prompt by LLM-as-a-Judge. The evaluation takes as input the question, the ground truth answer, and the final answer extracted from the model, and returns whether two answers are equivalent. We use `gpt-4.1-mini-2025-04-14` as the evaluation model, with a sampling temperature $t$=0.

