# OpenReview forum: "ChartMuseum: Testing Visual Reasoning Capabilities of Large Vision-Language Models"
_NeurIPS.cc/2025/Datasets_and_Benchmarks_Track — NeurIPS 2025 Datasets and Benchmarks Track poster_

### Official Review · Reviewer_VjAB · 2025-06-30

**Rating:** 5
**Confidence:** 4

**Summary:**

The paper introduces CHARTMUSEUM, a new chart-question-answering (QA) benchmark explicitly designed to probe the visual reasoning skills of large vision-language models (LVLMs). It couples this benchmark with a synthetic testbed and an extensive evaluation of 21 recent models, revealing a persistent gap between human and model performance—especially on questions that cannot be solved by text extraction alone. The authors release the dataset (1162 QA pairs over 928 real-world charts from 184 sources), the evaluation code, and an analysis taxonomy of the visual skills that today’s LVLMs lack.

**Dataset Code Accessibility:**

Yes

**Dataset Code Comments:**

The dataset is available on Hugging face.

**Ethical Considerations:**

No, there are no or only very minor ethics concerns

**Final Justification:**

The authors have addressed my concerns, and I saw the discussions between other reviewers and authors, I'm willing to accept this paper at this stage.

**Limitations Weaknesses:**

Accuracy is scored by GPT-4 as an “LLM-as-a-judge,” which risks favoring GPT-style answers and introduces circularity unless cross-checked with humans or alternative judges.

**Strengths Contributions:**

1. This benchmark focuses on true visual reasoning in chart QA—an area where existing benchmarks largely test text extraction, leaving plenty of headroom for progress.

2. All 1162 QA pairs are written and triple-checked by humans, balanced across four reasoning types, and paired with a clear error taxonomy—raising quality above auto-generated datasets.

3. Testing 21 LVLMs with a unified protocol exposes concrete failure modes (symbol selection, visual comparison, etc.) and a sizeable human-model gap, giving the community a ready-made leaderboard and roadmap for improvement.

---

> ### Author Rebuttal · Authors · 2025-07-30
>
> Thanks for your review and your comments on our work!
>
> > **Question 1: Accuracy is scored by GPT-4 as an “LLM-as-a-judge,” which risks favoring GPT-style answers and introduces circularity unless cross-checked with humans or alternative judges.**
>
> Please note that we are comparing **short answers** with LLM-as-a-judge here. The average length of the final short answer is 2.9 (Figure 6), which is usually an entity such as a name and a number. We believe this would not favor GPT-style answers. This follows standard recent practice in other datasets such as SimpleQA [1].
>
> During the human evaluation, we used both GPT-4.1 and humans to judge the human answers and their agreement on the 100-example human evaluation set is 100%.
>
> ___
> [1] Wei, Jason, et al. "Measuring short-form factuality in large language models." arXiv preprint arXiv:2411.04368 (2024).

---

> > ### Comment · Reviewer_VjAB · 2025-08-05
> >
> > Thanks for the rebuttal, I'm willing to maintain my score.

---

### Official Review · Reviewer_1Buo · 2025-07-02

**Rating:** 4
**Confidence:** 4

**Summary:**

This paper proposes a new benchmark for evaluating LVLMs' chart understanding abilities. The benchmark contains diverse (collected from various web sources) and human-curated question-answer pairs that pose challenge to state-of-the-art LVLMs. Extensive experimental results using existing LVLMs are presented, indicating the gap between human performance and machine performance. Additional analysis further suggests that the visual reasoning capability of LVLMs on chart images are far behind the textual understanding abilities.

**Additional Feedback:**

### Additional Questions
- What percentage of questions falls into each classification?
- How diverse is the overall question set?


### Comments
- Table 2 is somewhat missleading. ChartXiv are partially human written, so should be half check mark in Table 2?


### Final Justification
As a benchmark, I believe two aspects are critical:
(1) Clear Motivation:
The analysis finding that complex charts are challenging for LVLMs isn’t novel. Prior benchmarks like CharXiv and ChartBench already include such charts. Why the community needs an additional, similar dataset. Moreover, work such as Cambrian-1 [1] has already shown that LVLMs’ visual reasoning lags behind their textual reasoning (likely attributed to strong OCR performance). It’s unclear what unique insight this new benchmark provides.
(2) Data Quality and Diversity:
Although the authors emphasize fully human-curated question-answers, reviewers noted a bias toward “extreme value” questions (e.g., largest, lowest). Likewise, the dataset is dominated (around 50%) by simple bar, line, and scatter charts, which limits its representativeness. A more comprehensive analysis of question diversity (suggested under Weaknesses) is needed to validate the benchmark’s utility and ensure balanced coverage. Overall, considering the weak motivation and unclear data quality, my initial score is Reject. However, I'm willing to re-evaluate this work if the concerns in Weakness can be addressed.

[1] Tong, Peter, et al. "Cambrian-1: A fully open, vision-centric exploration of multimodal llms." Advances in Neural Information Processing Systems 37 (2024): 87310-87356.

**Dataset Code Accessibility:**

Yes

**Dataset Code Comments:**

Dataset and code are readily accessible.

**Ethical Considerations:**

No, there are no or only very minor ethics concerns

**Final Justification:**

I read the rebuttal and comments from other reviewers. The rebuttal and the discussion address most of my concerns. Although I still have a few concerns listed below regarding the writing in the analysis section, I believe that, at this moment, these concerns are primarily related to writing issues that can be addressed in the camera-ready version. I will increase my initial score, and I encourage the authors to revise the writing concerns accordingly in the camera-ready version.

Suggestions to improve:

- Clarify the textual extraction task. Explicitly mention that the textual extraction refers to OCR detection, and if the charts are non-annotated, the underlying data won't be extracted.
- Clarify the relationship between visual reasoning and chart complexity.
- Include the analysis results of Question Diversity in the paper.
- Include the comparison of chart-type distribution between CharXiv and ChartMuseum.
- Include the missing citations and discussion of previous works (e.g., Cambrian-1) in the related works section of the paper.

**Limitations Weaknesses:**

### Major concerns
- Question Diversity: The benchmark’s questions are formualted by human annotators under two constraints: a large answer space and answer objectivity. However, there are no guidance to ensure question diversity across annotators. As a result, annotators tend to focus on extreme chart values (e.g., largest, lowest, rightmost, leftmost). After manually reviewing ~100 test samples on hugginface, I found that many questions indeed target extreme values. I suggest conducting a formal diversity analysis to quantify and mitigate this bias.

- Human-only vs Machine-augmented Questions: Given the lack of question diversity, it’s unclear why fully human-curated questions are preferred over a hybrid approach (as used in CharXiv). In CharXiv's workflow, GPT first generates a diverse pool of template questions, explicitly prompted to cover various aspects, and then humans pick up and refine them to reduce annotator bias. CharXiv’s questions span not only information extraction but also higher-order reasoning tasks (e.g., mean, median, max, standard deviation, comparisons, trend analysis). The reviewer is concern about whether such complex reasoning questions would naturally formed from human annotators, given the extra effort required.

- Synthetic Data Analysis: The synthetic data results show that charts with greater visual complexity (i.e., overlapping charts) are harder for LVLMs. This is an unsurprising finding that offers little new insight. The motivation for introducing a new benchmark should be better formulated beyond this obvious observation.

### Minor concerns
- Dataset Size: With only 1,000 samples overall, and as few as 100 for certain subtasks, I'm worried about that the benchmark may suffer from high variance and limited statistical power.

- Chart-type Distribution: ~50% of the charts are bar, line, or scatter plots. In contrast, CharXiv presents a more balanced mix of chart types (as shown in the random sample qualitative results). Although advanced types (e.g., radar, heatmap, treemap) are included in the set, each appears in only ~20 samples, limiting the ability to evaluate model performance on them.

- Clarity of Table 1: What is the core message that authors want to deliever? Table 1 shows that ChartMuseum’s performance using extracted data is much lower than using the raw images. Is this due to errors in the data extraction process, or because the model struggles to reason effectively even when the extraction is correct? Please provide more details explanation.

**Strengths Contributions:**

- The paper is well-written and easy to follow.

- The dataset and code are well-documented and readily accessible.

- Extensive experimental results from various LVLMs are provided, and the results are interesting.

---

> ### Author Rebuttal · Authors · 2025-07-31
>
> Thanks for your review and your comments on our work!
>
> > **Question 1: Question Diversity: [...] Annotators tend to focus on extreme chart values… After manually reviewing ~100 test samples, I found that many questions indeed target extreme values.**
>
> We appreciate you taking such a close look at our data! Based on your comment, we conducted our own analysis of question diversity for two datasets: ChartMuseum and the reasoning subset of CharXiv.  We randomly sampled 50 questions from each and manually examined how many targeted extreme values. The results showed that 25/50 questions in CharXiv and 26/50 in ChartMuseum focused on extremes, with no statistically significant difference between the two.
>
> *Of the sampled 50 questions from CharXiv, the questions share similar reasoning required to answer the questions, but mainly differ by the fact that the questions are based on different chart images (see candidate questions in Figure 10 of the CharXiv paper).* We believe this is the fundamental limitation of creating question templates and letting models generate similar questions based on the template.
>
> In ChartMuseum, we directly ask humans to independently write questions with no shared prior. Even if many questions are about extreme values, the questions are not direct lookups, and require strong **multi-step reasoning capability unique to each image**. A few representative examples from the sampled 50 examples from ChartMuseum include:
> * *(images/Tour de France.png) In 1980s, which year has the highest proportion of the race not led by the eventual winner?*
>
>   The question requires a deep understanding of the context of the chart and fine-grained comparison of over 100 subplots.
> * *(images/tennis_scores_women.png) Which player has the smallest variance in the percentages won across points, games, sets, and matches?*
>
>   The question requires a model to visually compare the variance of each player’s performance
>
> In short, **we found that our sample extreme value questions required more unique visual reasoning skills to answer compared to those extreme value questions in CharXiv.**
>
> > **Question 2: Human-only vs Machine-augmented Questions: it’s unclear why fully human-curated questions are preferred over a hybrid approach (as used in CharXiv).**
>
> At an early stage of the project, we explored the use of templated questions to help with annotation, but were not satisfied with the distinctiveness or quality of generated questions. We aim to generate questions that are **unique** to each different chart. When not using templates, our annotators generally pose questions targeting the specific core messages that the chart aims to convey, which requires understanding and interpretation of the chart.
>
> > **Question 3: CharXiv’s questions span not only information extraction but also higher-order reasoning tasks [...] The reviewer is concerned about whether such complex reasoning questions would naturally form from human annotators, given the extra effort required.**
>
> Many questions from ChartMuseum require multi-step higher-order reasoning. Examples:
> * **(Mean)** What is the average global per capita production in 2022 (in kilograms) across the types of food whose global per capita production in 1961 is exactly the same as its production in 2022? (images/global_food_production.png)
> * **(Comparison)** Which part of latency (among the three listed in legend) is the most optimized switching from Naive to Ours across different models? (images/latency_apple.png)
>
> > **Question 4: Synthetic Data Analysis: The synthetic data results show that charts with greater visual complexity (i.e., overlapping charts) are harder for LVLMs. This is an unsurprising finding that offers little new insight.**
>
> Thanks for pointing this out! Our motivation in this experiment is to **contrast between text reasoning and visual reasoning**, in addition to showing the scaling along the visual complexity. **We are the first to show that current chart understanding benchmarks such as ChartQA can be mostly done by a text-only LLM**. We want to highlight this insight as we found that *many recent papers are still using ChartQA dataset as the main benchmark to test “visual” reasoning on chart understanding* as of July 2025 [1,2,3].
>
> [1] Team, Kwai Keye, et al. "Kwai Keye-VL Technical Report." arXiv preprint arXiv:2507.01949 (2025).
>
> [2] Fan, Wan-Cyuan, et al. "In-Depth and In-Breadth: Pre-training Multimodal Language Models Customized for Comprehensive Chart Understanding." arXiv preprint arXiv:2507.14298 (2025).
>
> [3] Luo, Gen, et al. "Mono-InternVL-1.5: Towards Cheaper and Faster Monolithic Multimodal Large Language Models." arXiv preprint arXiv:2507.12566 (2025).
>
> > **Question 5: Dataset Size: With only 1,000 samples overall, and as few as 100 for certain subtasks, I'm worried that the benchmark may suffer from high variance and limited statistical power.**
>
> Thanks for the suggestion. We conducted a paired bootstrap test (10k times) on the performance difference between the best model Gemini-2.5-Pro and all other models on the overall benchmark and 4 reasoning subsets. We found that Gemini-2.5-Pro outperformed every model for which the absolute difference was >3% with  p-value < 0.05. In general, differences of about 3-4% on the dataset hold up under bootstrap tests, making it large enough to differentiate practically meaningful differences in model performance.
>
> > **Question 6: Chart-type Distribution: ~50% of the charts are bar, line, or scatter plots. In contrast, CharXiv presents a more balanced mix of chart types.**
>
> To make a fair comparison, we randomly sampled 50 arXiv charts from ChartMuseum and CharXiv, respectively, and manually checked the chart type distributions as shown below:
>
> ChartMuseum – arXiv Charts (percentage of charts with subplots: 58%)
>
> | Type | Percentage |
> |------|------------|
> | Line | 26% |
> | Bar/Histogram | 22% |
> | Scatter | 12% |
> | Multiple chart types | 16% |
> | Box plot | 6% |
> | Heatmap | 2% |
> | Flowchart | 4% |
> | Area | 6% |
> | 3D chart | 2% |
> | Density | 2% |
> | Radar | 2% |
>
>
> ChartXiv (reasoning set) – arXiv Charts  (percentage of charts with subplots: 54%)
>
> | Type | Percentage |
> |------|------------|
> | Line | 40% |
> | Bar/Histogram | 14% |
> | Scatter | 16% |
> | Multiple chart types | 16% |
> | Heatmap | 14% |
>
>
> Although we do not include as many arXiv chart images in the paper Appendix as in CharXiv, ChartMuseum does have a diverse mix of chart types for the arXiv charts.
>
> On the other hand, even if the chart types can be simple as bar charts, a chart can still be visually diverse and challenging to answer, especially in the infographic-style charts. For instance,
> * (**Line Chart**) List all months that have never been the ""month with the highest average birth rate per day"" (displayed as the dots in the chart) for any of the years from 1861 to 2022? (images/common_birth_month.png)
>
> > **Question 7: Clarity of Table 1: What is the core message that authors want to deliver?**
>
> The large performance gap by only using image or extracted text to answer questions in ChartMuseum reflects our intent to design questions that inherently require visual reasoning, which cannot be captured in extracted text. For both datasets, LVLMs such as Claude 3.7 can extract factually-correct information from the charts. However, the extracted data itself is not sufficient to answer the questions in ChartMuseum. For instance, it might not cover a sufficient amount of data in a chart to reflect an overall trend.
>
> > **Question 8: How diverse is the overall question set?**
>
> Please refer to our responses to question 1, 2, 3 and 6.
>
> > **Question 9: Table 2 is somewhat misleading. ChartXiv are partially human written, so should be half check mark in Table 2?**
>
> Thanks for spotting the error! We will correct it in the future version.
>
> > **Question 10: Need Clear Motivation: The analysis finding that complex charts are challenging for LVLMs isn’t novel. Prior benchmarks like CharXiv and ChartBench already include such charts. Why the community needs an additional, similar dataset.**
>
> Our main motivation is to differentiate visual reasoning and textual reasoning. Prior work only differentiates descriptive and reasoning questions, but we further divided reasoning questions into textual and visual reasonings and emphasized that current benchmarks mostly target textual reasoning questions (such as ChartQA).  CharXiv and ChartBench target the visual dimension to some extent, but CharXiv is domain-limited and ChartBench mostly focuses on visual perception and does not contain complex visual reasoning charts and questions. In addition, images from ChartBench are synthetically generated from model generated code.
>
> > **Question 11: Work such as Cambrian-1 has already shown that LVLMs’ visual reasoning lags behind their textual reasoning. It’s unclear what unique insight this new benchmark provides.**
>
> Cambrian-1 revealed general findings, but we are providing unique insights on chart understanding. We summarize the unique insights that ChartMuseum provides as the followings:
> 1. (Section 4.2) We explicitly divided our benchmark into subsets based on the textual/visual reasoning skills. On questions requiring primarily visual reasoning, all models experience a 35%-55% performance drop from text-reasoning-heavy question performance. **The benchmark provides specific targets for improvement** (e.g. humans achieve 98.2% on visual reasoning questions) rather than just knowing "there's a gap.”
> 2. (Section 4.2) We found that **extended-thinking does not help in chart understanding that requires complex visual reasoning**.
> 3. (Section 5.1) We have **a detailed analysis of visual reasoning failures on the SOTA models**, which provides insights into the visual reasoning dimensions that future models could improve on.
> 4. (Section 5.2) We found that **current LVLMs actively avoid visual reasoning strategies and over-rely on textual reasoning strategies**.

---

> > ### Comment · Reviewer_1Buo · 2025-08-05
> >
> > I appreciate the response from the authors. The rebuttal addresses some of my initial concerns, but a few issues remain. Here are the details:
> >
> > > Question 7: Clarity of Table 1: What is the core message that authors want to deliver?
> >
> > I’m still unclear about the analysis in Tab. 1. The authors state that "LVLMs such as Claude 3.7 can extract **factually-correct information from the charts**" for both datasets. Appendix Figure 24 shows that the extracted text includes titles, axis labels, legends, and raw data. Yet Tab. 1 reveals a 40% performance gap between using only the extracted text and using only the image. This suggests that Claude, even with all chart data correctly captured as text, still fails to answer the questions correctly, pointing more to a textual reasoning limitation than a visual reasoning issue. I’m not sure (1) why the model performs worse even with the chart information fully available as text, and (2) how this gap reflects a lack of visual reasoning when it seems driven by a lack of textual reasoning.
> >
> > > Question 4: Synthetic Data Analysis
> >
> > The authors explain that "this experiment contrasts text reasoning and visual reasoning and shows how performance scales with visual complexity. We are the first to show that benchmarks like ChartQA can be mostly solved by a text-only LLM." However, I’m confused how the synthetic data analysis in Fig. 2 supports the claim that **ChartQA can be solved with text alone**. My understanding was that this experiment probes visual reasoning: we observe that more complex charts lead to lower LVLM performance. But I still not sure how this finding connects to visual reasoning, nor what the main takeaway of this analysis is.
> >
> > It will be appreciated if the authors can provide further discussion.

---

> > > ### Author Response · Authors · 2025-08-06
> > >
> > > Thanks for bringing up those clarification questions!
> > >
> > > > **Question 7**
> > >
> > > The key emphasis here is that complete factually-correct text extraction is not equivalent to sufficient information for reasoning. When we refer to text extraction, we refer to the behavior of LLMs (including strong models like Claude 3.7) which accurately extract **explicit** textual elements such as titles, axis labels, legends, and annotated values (e.g., text labels on bar graphs).  These LLMs **do not** extract a complete readout of the data series from the chart; for instance, if given a scatterplot with hundreds of points (images/cot_or_not_fig1a.png), they won’t enumerate all distinct values and cannot be easily prompted to do so. The 40% performance gap in ChartMuseum *reflects information that is inherently visual and cannot be captured in the **easily-accessible text form**, which is not a textual reasoning limitation*. **Our human study finds that even if the extracted text is accurate and complete in ChartMuseum, humans cannot get sufficient information to answer the question with only the extracted text.**
> > >
> > > **Many ChartMuseum questions require reasoning about visual properties that cannot be easily converted to text.** For example, if a question asks about which of clusters in a scatterplot is the most concentrated (images/player_metres_tackles.png), **this information cannot be conveyed through explicitly extracted text produced by LLMs.** Theoretically, an LLM could attempt to read off every single data value from the scatterplot, but LLMs don’t do this, especially when LLMs extract the explicit text before seeing the questions. When we extract all available textual information from a chart (explicit labels) and other broad descriptions of trends, we inevitably miss crucial visual information needed for reasoning.
> > >
> > > On the other hand, questions from ChartQA can typically be answered using the explicit textual elements that are extractable, such as exact values and labels. The 40% performance gap thus reflects visual reasoning requirements, not textual reasoning limitations. This gap demonstrates that our benchmark successfully requires visual understanding that cannot be reduced to explicit textual processing, which was our core design goal.
> > >
> > > > **Question 4**
> > >
> > > You're right that the synthetic data analysis in Figure 2 and the ChartQA text-extraction analysis in Table 1 are separate experiments that serve different purposes.
> > >
> > > The ChartQA text-extraction analysis was designed to show that existing benchmarks can be solved primarily through textual reasoning. In contrast, our synthetic data analysis probes pure visual reasoning capabilities when text extraction is impossible. We created synthetic charts with no text labels that require visual interpretation (i.e., it would be very hard to extract them into concise text representations without visual reasoning), and found that LVLM performance is worse with higher visual complexity .
> > >
> > > **The main takeaway of synthetic analysis is that when LVLMs cannot rely on text extraction shortcuts, their visual reasoning capabilities are severely limited and degrade with complexity**. Together, these two experiments highlight an existing evaluation gap: the synthetic analysis shows LVLMs struggle with pure visual reasoning, while the ChartQA analysis shows existing benchmarks don't require such reasoning. This gap is precisely what ChartMuseum addresses by providing a benchmark that requires visual reasoning in real-world contexts.

---

> > > > ### Comment · Reviewer_1Buo · 2025-08-08
> > > >
> > > > I appreciate the comment and further clarification from the authors. The rebuttal and the discussion address most of my concerns. Although I still have a few concerns listed below regarding the writing in the analysis section, I believe that, at this moment, these concerns are primarily related to writing issues that can be addressed in the camera-ready version. I will increase my initial score, and I encourage the authors to revise the writing concerns accordingly in the camera-ready version.
> > > >
> > > > Suggestions to improve:
> > > >
> > > > - Clarify the textual extraction task. Explicitly mention that the textual extraction refers to OCR detection, and if the charts are non-annotated, the underlying data won't be extracted.
> > > > - Clarify the relationship between visual reasoning and chart complexity.
> > > > - Include the analysis results of Question Diversity in the paper.
> > > > - Include the comparison of chart-type distribution between CharXiv and ChartMuseum.
> > > > - Include the missing citations and discussion of previous works (e.g., Cambrian-1) in the related works section of the paper.

---

> > > > > ### Author Response · Authors · 2025-08-09
> > > > >
> > > > > We really appreciate your careful and thorough review and will address those writing issues in the camera-ready version! We also wanted to kindly mention that the rating field may have been unintentionally missed.

---

### Official Review · Reviewer_gnoN · 2025-07-03

**Rating:** 4
**Confidence:** 3

**Summary:**

This work proposes to evaluate visual reasoning ability of VLMs. They first conduct a case study using 100 synthetic chart data and evaluate using Claude-3.7-Sonnet. The results indicate the visual complexity decreases the model performance, but humans are not influenced significantly. Then, they propose CHART-MUSEUM, a Chart QA benchmark including 1162 questions on real world charts from 184 sources, created purely by humans (researchers). The questions are classified into pre-defined categories (e.g., visual reasoning questions/textual reasoning questions). The quality of the annotation is checked by iteratively refining questions and answers. They evaluate on 21 open-sourced and proprietary VLMs. The results show that visual reasoning is more difficult than textual reasoning for current VLMs, and human can achieve higher accuracy than any of the models. The authors also analyze the errors by visual task taxonomy and show that models such as Claude-3.7 and Gemini-2.5-pro cannot do well in symbol selection and visual comparison instances obviously.

**Dataset Code Accessibility:**

Yes

**Dataset Code Comments:**

The data is provided

**Ethical Considerations:**

No, there are no or only very minor ethics concerns

**Final Justification:**

The rebuttal analyzes the visual encoders and the data used for visual reasoning. If the authors can adapt them in the final version I think I don't have more concerns and will keep my rating then.

**Limitations Weaknesses:**

(1) When working on visual reasoning, the influences of visual encoders should be considered as well, and the authors may use settings of visual encoders (e.g., size, model type) to further analyze connection between VLMs and visual reasoning ability.

(2) The training data may also influence the performances on visual reasoning. The authors can also consider to analyze the open-sourced model performances based on the data information.

**Strengths Contributions:**

(1) Visual reasoning is important for current VLMs, since VLMs usually rely a lot on LLMs, but the visual components are important in real life scenario.

(2) The paper is well-written and easy to follow.

(3) The proposed benchmark provides high-quality QA pairs and the categories can help the research society for evaluating VLMs in a more detailed setting.

---

> ### Author Rebuttal · Authors · 2025-07-30
>
> Thanks for your review and your comments on our work!
>
> > **Question 1: When working on visual reasoning, the influences of visual encoders should be considered as well, and the authors may use settings of visual encoders (e.g., size, model type) to further analyze connection between VLMs and visual reasoning ability.**
>
> Thanks for the great question! Since closed-source models do not reveal details about their visual encoders, we use the best open-source models from the Qwen2.5-VL family for this analysis. According to the technical report of Qwen2.5-VL [1], all models in the Qwen2.5-VL family (3B, 7B, 72B) use identical visual encoders (Table 1 in [1]), with the only difference being the size of their language models.
>
> Our results on ChartMuseum reveal a clear scaling effect: despite using the same visual encoder, the 72B model significantly outperforms the 7B model (Table 3). This suggests that while the visual encoder extracts the same visual information across model sizes, **larger LLMs are more effective at decoding and reasoning over these visual features**. This finding aligns with [2] and indicates that visual reasoning improvements come from both better visual feature decoding and enhanced reasoning capabilities over extracted information.
>
> The performance of chart-specialized models provides additional insights here. The Bespoke-MiniChart-7B model (34.0%), which is specifically optimized for chart understanding, outperforms the general-purpose Qwen2.5-VL-7B (26.8%) and approaches Qwen2.5-VL-72B (38.5%). This shows that **proper training can enable smaller models to decode visual features more effectively than off-the-shelf models**, suggesting that both model size and task-specific optimization lead to better visual reasoning performance.
>
> However, we observe a potential bottleneck in current visual encoders. The performance gap between Qwen2.5-VL-32B (38.1%) and Qwen2.5-VL-72B (38.5%) is minimal, suggesting that **visual encoder limitations may impose an upper bound on performance regardless of LLM size**. This indicates that further scaling language model parameters alone may not yield proportional improvements in visual reasoning tasks.
>
> Finally, our analysis of extended reasoning (Section 4.2) shows that longer textual CoT provides limited benefits for visual feature decoding. The marginal improvement from Claude-3.7 (60.3%) to Claude-3.7 with extended thinking (61.7%) suggests that **visual reasoning cannot be fundamentally improved through purely textual reasoning extensions**.
>
> We will include this discussion in any future version!
>
> > **Question 2: The training data may also influence the performances on visual reasoning. The authors can also consider analyzing the open-sourced model performances based on the data information.**
>
> According to the technical report of Qwen2.5-VL [1], there are various data related factors that can affect visual reasoning performance, such as scaling of pretraining visual data, filtering of high-quality data, and collecting CoT reasoning traces.
>
> In addition, our results demonstrate that **training on specialized data can significantly compensate for model size limitations**. The chart-specialized Bespoke-MiniChart-7B (34.0%) substantially outperforms the general-purpose Qwen2.5-VL-7B (26.8%) despite having identical model architecture and size. This specialized 7B model also approaches the performance of the much larger Qwen2.5-VL-72B (38.5%), suggesting that targeted training on chart-specific data can be as effective as scaling model parameters by an order of magnitude.
>
> ___
>
> [1] Bai, Shuai, et al. "Qwen2. 5-vl technical report." arXiv preprint arXiv:2502.13923 (2025).
>
> [2] Huang, Kung-Hsiang, et al. "Why vision language models struggle with visual arithmetic? towards enhanced chart and geometry understanding." arXiv preprint arXiv:2502.11492 (2025).

---

> > ### Comment · Reviewer_gnoN · 2025-08-06
> > **thanks**
> >
> > Thank the authors for providing the additional analysis. I don't have more concerns at this time.

---

### Official Review · Reviewer_RrVs · 2025-07-04

**Rating:** 5
**Confidence:** 4

**Summary:**

This work proposes a new benchmark -- ChartMuseum to evaluate the textual and visual reasoning capabilities of VLMs in the Chart QA task. It consists of around 1k human-curated QA pairs from over 900 real-world charts collected from 184 sources spanning diverse domains and chart types. The benchmark addresses a critical gap in current evaluation practices by focusing on visual reasoning tasks that require more than just simple textual extraction.

The authors evaluate 21 VLMs, including both proprietary and open-source models, across four reasoning categories: textual, visual, visual/textual, and synthesis. The results show that while SOTA models excel in textual reasoning, they fall short in visual reasoning tasks by 35-55%. Humans outperform the best model -- Gemini-2.5-Pro (93% vs 63%) on this test set. Additionally, the authors introduce a synthetic visual reasoning dataset to further isolate the skill gap and provide a more detailed qualitative error analysis, categorizing visual challenges into symbol selection, trajectory tracking, and visual comparison.

**Additional Feedback:**

How was the final reasoning-type label for each question determined? Was it based solely on the annotator’s judgment or validated by additional reviewers?


**Corrections:**

line 25 -- citation missing

**Dataset Code Accessibility:**

Yes

**Ethical Considerations:**

No, there are no or only very minor ethics concerns

**Final Justification:**

Most of the concerns are addressed by the authors in the rebuttal. I look forward to these being incorporated into the final manuscript.

**Limitations Weaknesses:**

1. Most images in the benchmark are associated with only one QA pair. I think that this limits the potential of the benchmark to probe multi-faceted reasoning about a single chart and prevents investigating how related queries might be answered and evaluating model consistency. What is the rationale for not allowing multiple questions per chart to explore different reasoning types?

2. The evaluation solely focuses on short-answer QA and excludes instances that involve reasoning types such as summarization, explanation, or multi-step inference across multiple charts. While this use case is one of the most widely used when interfacing these models with users, it is not currently covered in the benchmark. How do you think this benchmark can be extended for longer-form tasks like visual summarization and so on?

3. The benchmark is currently limited to charts and QA pairs in the English language. However, this limitation might restrict its applicability for assessing multilingual or cross-lingual Chart QA tasks, where layout and annotation styles may differ.

**Strengths Contributions:**

1. Existing ChartQA benchmarks rely on LLMs for question generation, which could introduce ambiguity and subjectivity into the test set, leading to noisy evaluation results. On the other hand, this benchmark is entirely human-curated, avoiding the use of LLMs for question generation or verification. It employs rigorous annotation guidelines and a multi-stage review process to ensure high-quality and diverse QA pairs.

2. The authors in this work explicitly categorize the reasoning types for each question into textual and visual, allowing for more nuanced model evaluation and error analysis. This granularity offers deeper insight into the specific reasoning skills that models lack and will motivate future research to address these shortcomings.

3. The authors evaluate 21 models, including proprietary and open-source models, to identify potential gaps in skills capabilities between these models. This significant gap between textual and visual reasoning types shows the effectiveness of the benchmark in addressing the issue of near saturation in existing benchmarks.

4. Unlike existing benchmarks, ChartMuseum includes real-world charts from a wide range of sources rather than synthetic or academic-only domains. This evaluation offers a more realistic assessment of the capabilities of VLMs and also makes it useful to determine the practical applicability of VLMs in real-world use cases.

5. The experiments conducted on the synthetic subset further highlight limitations of these models in visual reasoning, particularly in a controlled setting across different categorizations. The analysis of this subset isolates vision-specific deficiencies in the current models.

---

> ### Author Rebuttal · Authors · 2025-07-30
>
> Thanks for your review and your comments on our work!
>
> > **Question 1: Most images in the benchmark are associated with only one QA pair. [...] What is the rationale for not allowing multiple questions per chart to explore different reasoning types?**
>
> Given the annotation budget, we are optimizing for question and image diversity at the expense of more related questions per image. Furthermore, we found that it was often challenging to come up with natural additional questions for each image that would also have high question difficulty while requiring a totally different solution strategy. The questions we devise are often related to the core information the visualization is meant to convey, and many infographics have a single core message. Although it is relatively easy to come up with similar questions by simply changing the entities, we felt it was better to focus towards generating one challenging question per chart and aim for a higher number of different charts.
>
> > **Question 2: The evaluation solely focuses on short-answer QA and excludes instances that involve reasoning types such as summarization, explanation, or multi-step inference across multiple charts. While this use case is one of the most widely used when interfacing these models with users, it is not currently covered in the benchmark. How do you think this benchmark can be extended for longer-form tasks like visual summarization and so on?**
>
> Although our benchmark does not contain multi-image instances, *we do have over 20% charts that inherently contain subplots (>50% for arXiv charts) and require multi-step inference*. We didn’t include subplot examples in the appendix as those charts with subplots are usually large images. We can include several subplot examples in the leaderboard of the paper or in a future version.
>
> As mentioned in the limitations, we believe short-answer QA is an effective proxy for identifying model weaknesses, as other tasks can often be reformulated as QA or are inherently subjective to evaluate (e.g., summarization). We found that without specifically asking about the facts in the chart, the LVLM usually will only mention high-level trends and findings of the chart, which are usually factually correct but debatable whether they really capture the main idea. However, this is difficult to capture reliably with current automated evaluation paradigms.
>
> > **Question 3: The benchmark is currently limited to charts and QA pairs in the English language. However, this limitation might restrict its applicability for assessing multilingual or cross-lingual Chart QA tasks, where layout and annotation styles may differ.**
>
> We agree, and discuss this in our limitation section. Overall, since most current LVLMs are optimized for English, we believe our benchmark provides a timely evaluation of their capabilities without introducing an extra dimension of linguistic competency in other languages, but we agree it would be great to extend to other languages in the future!
>
> > **Question 4: How was the final reasoning-type label for each question determined? Was it based solely on the annotator’s judgment or validated by additional reviewers?**
>
> As mentioned in section 3.2, every stage of the benchmark annotations requires (1) an independent check from an independent reviewer (the first author of this work, who was not among the 13 annotators); (2) and a consensus between the annotators and the independent reviewer is required for any differences, including the reasoning-type labels.
>
> > **Question 5: line 25 -- citation missing**
>
> Thanks and we will fix this in the future version!

---

> > ### Comment · Reviewer_RrVs · 2025-08-06
> > **Response to author rebuttal**
> >
> > I appreciate the authors' effort for the response and it indeed addresses some of the concerns. I do agree that short-answer QA provides a more deterministic way of evaluating these models on Charts, but I also think this fails to capture a more realistic assessment of the chart understanding skills in LVLMs, where the ground truths may not always be shorter.
> >
> > However, after reading the points raised by the other reviewers, I also kind of agree with some points raised by reviewer 1Buo. For instance, Q7 from reviewer 1Buo highlights that the primary limitation appears to lie in textual reasoning rather than visual reasoning, as per the experiments on the Claude 3.7 model reported in Tab 1 -- where textual information is extracted from the chart and directly included in the prompt. I would appreciate further clarification from the authors on this point. It is important that the scope and claims of the paper accurately reflect these findings, as they suggest a different finding than the one currently presented.
> >
> > I prefer to retain my rating as before and believe that it already reflects the merits of the work. Thank you!

---

> > > ### Author Response · Authors · 2025-08-06
> > >
> > > Thanks for your careful review of our rebuttal! Here is our response regarding to "textual reasoning limitations."
> > >
> > > The key emphasis here is that complete factually-correct text extraction is not equivalent to sufficient information for reasoning. When we refer to text extraction, we refer to the behavior of LLMs (including strong models like Claude 3.7) which accurately extract **explicit** textual elements such as titles, axis labels, legends, and annotated values (e.g., text labels on bar graphs).  These LLMs **do not** extract a complete readout of the data series from the chart; for instance, if given a scatterplot with hundreds of points (images/cot_or_not_fig1a.png), they won’t enumerate all distinct values and cannot be easily prompted to do so. The 40% performance gap in ChartMuseum *reflects information that is inherently visual and cannot be captured in the **easily-accessible text form**, which is not a textual reasoning limitation*. **Our human study finds that even if the extracted text is accurate and complete in ChartMuseum, humans cannot get sufficient information to answer the question with only the extracted text.**
> > >
> > > **Many ChartMuseum questions require reasoning about visual properties that cannot be easily converted to text.** For example, if a question asks about which of clusters in a scatterplot is the most concentrated (images/player_metres_tackles.png), **this information cannot be conveyed through explicitly extracted text produced by LLMs.** Theoretically, an LLM could attempt to read off every single data value from the scatterplot, but LLMs don’t do this, especially when LLMs extract the explicit text before seeing the questions. When we extract all available textual information from a chart (explicit labels) and other broad descriptions of trends, we inevitably miss crucial visual information needed for reasoning.
> > >
> > > On the other hand, questions from ChartQA can typically be answered using the explicit textual elements that are extractable, such as exact values and labels. The 40% performance gap thus reflects visual reasoning requirements, not textual reasoning limitations. This gap demonstrates that our benchmark successfully requires visual understanding that cannot be reduced to explicit textual processing, which was our core design goal.

---

> > > > ### Comment · Reviewer_RrVs · 2025-08-07
> > > >
> > > > Thanks for the clarification about textual reasoning vs visual reasoning limitations. I don't have any other concerns as of now and I look forward to the changes being incorportated in the revised manuscript.

---

### Decision · Program_Chairs · 2025-09-18

**Decision:**

Accept (poster)

**Comment:**

This work proposes a new benchmark -- ChartMuseum to evaluate the textual and visual reasoning capabilities of VLMs in the Chart QA task. It consists of around 1k human-curated QA pairs from over 900 real-world charts collected from 184 sources spanning diverse domains and chart types. The benchmark addresses a critical gap in current evaluation practices by focusing on visual reasoning tasks that require more than just simple textual extraction.
The authors evaluate 21 VLMs, including both proprietary and open-source models, across four reasoning categories: textual, visual, visual/textual, and synthesis. The results and analysis suggests that the visual reasoning capability of LVLMs on chart images are far behind the textual understanding abilities: while SOTA models excel in textual reasoning, they fall short in visual reasoning tasks by 35-55%. Humans outperform the best model -- Gemini-2.5-Pro (93% vs 63%) on this test set.
Additionally, the authors introduce a synthetic visual reasoning dataset to further isolate the skill gap and provide a more detailed qualitative error analysis, categorizing visual challenges into symbol selection, trajectory tracking, and visual comparison.